# Beyond Log-Concavity and Score Regularity: Improved Convergence Bounds for Score-Based Generative Models in $\mathcal{W}_2$-distance

**Marta Gentiloni-Silveri** [* 1]   **Antonio Ocello** [* 1]

## Abstract

Score-based Generative Models (SGMs) aim to sample from a target distribution by learning score functions using samples perturbed by Gaussian noise. Existing convergence bounds for SGMs in the $\mathcal{W}_2$-distance rely on stringent assumptions about the data distribution. In this work, we present a novel framework for analyzing $\mathcal{W}_2$-convergence in SGMs, significantly relaxing traditional assumptions such as log-concavity and score regularity. Leveraging the regularization properties of the Ornstein–Uhlenbeck (OU) process, we show that weak log-concavity of the data distribution evolves into log-concavity over time. This transition is rigorously quantified through a PDE-based analysis of the Hamilton–Jacobi–Bellman equation governing the log-density of the forward process. Moreover, we establish that the drift of the time-reversed OU process alternates between contractive and non-contractive regimes, reflecting the dynamics of concavity. Our approach circumvents the need for stringent regularity conditions on the score function and its estimators, relying instead on milder, more practical assumptions. We demonstrate the wide applicability of this framework through explicit computations on Gaussian mixture models, illustrating its versatility and potential for broader classes of data distributions.

## 1. Introduction

In recent years, machine learning has made remarkable progress in generating samples from high-dimensional distributions with intricate structures. Among various genera-

tive models, *Score-based Generative Models* (SGMs) (see, *e.g.*, Sohl-Dickstein et al., 2015; Song & Ermon, 2019; Ho et al., 2020; Song et al., 2020b;a) have gained significant attention due to their versatility and computational efficiency. These models leverage the innovative approach of reversing the flow of a Stochastic Differential Equation (SDE) and employ advanced techniques for learning time-reversed processes, enabling the generation of high-quality and visually compelling samples (Ramesh et al., 2022).

This breakthrough has spurred a rapid expansion in the applications of SGMs, which now span diverse domains. Examples include natural language generation (Gong et al., 2022), imputation for missing data (Zhang et al., 2024b), computer vision tasks such as super-resolution and inpainting (Li et al., 2022; Lugmayr et al., 2022), and even applications in cardiology (Cardoso et al., 2023). For a comprehensive review of recent advancements, we refer readers to Yang et al. (2023).

**SGMs.** The primary goal of these models is to generate synthetic data that closely match a target distribution $\pi_{\text{data}}$, given a sample set. This is particularly valuable when the data distribution is too complex to be captured by traditional parametric methods. In such scenarios, classical maximum likelihood approaches become impractical, and non-parametric techniques like kernel smoothing fail due to the high dimensionality of real-world data.

SGMs address these challenges by focusing on the *score function*—the gradient of the log-density of the data distribution—instead of directly modeling the density itself. The score function describes how the probability density changes across different directions in the data space, guiding the data generation process.

The generation process begins with random noise and iteratively refines it into meaningful samples, such as images or sounds, through a denoising process that reverses a forward diffusion process. The forward diffusion process starts with a data sample (*e.g.*, an image) and gradually corrupts it by adding noise over several steps through an SDE flow, eventually transforming it into pure noise.

During generation, SGMs learn to reverse this diffusion process. Using noisy versions of the data, the model trains a deep neural network to approximate the score function.

---

[*]Equal contribution   [1]CMAP, CNRS, École polytechnique, Institut Polytechnique de Paris, 91120 Palaiseau, France. Correspondence to: Marta Gentiloni-Silveri <marta.gentiloni-silveri@polytechnique.edu>, Antonio Ocello .

*Proceedings of the 42nd International Conference on Machine Learning*, Vancouver, Canada. PMLR 267, 2025. Copyright 2025 by the author(s).

Starting from random noise, the model applies the learned score function iteratively, progressively removing noise and refining the sample to match the original data distribution. At each step, small, controlled adjustments based on the score function guide the noise toward realistic data, ensuring that the final sample aligns closely with the target distribution.

**HJB approach.** Connections between SGMs and partial differential equation (PDE) theory have been explored in recent literature through various interpretations. For instance, Berner et al. (2022) demonstrates that the log-densities of the marginal distributions of the underlying SDE satisfy a specific Hamilton–Jacobi–Bellman (HJB) equation. By applying methods from optimal control theory, they reformulate diffusion-based generative modeling as the minimization of the KL divergence between appropriate measures in path space. This perspective has been further developed in works such as Zhang & Katsoulakis (2023); Zhang et al. (2024a), where SGMs are viewed as solutions to a mean-field game problem, coupling an HJB equation—representing an infinite-dimensional optimization—with the evolution of probability densities governed by the Fokker–Planck equation.

A connection between time-reversal and stochastic control has been identified in Cattiaux et al. (2023); Conforti & Léonard (2022), providing a powerful analytical perspective for studying diffusion processes. This framework, with the associated HJB equation, has been effectively leveraged in recent works such as Conforti et al. (2023a); Pham et al. (2025), where the authors utilize it to derive tight convergence bounds in KL divergence for score-based generative models, under minimal regularity assumptions on the data distribution.

However, convergence bounds in terms of the Wasserstein distance through coupling methods (Villani, 2021) and the study of the associated HJB equation remains an open problem. Among the various coupling techniques proposed, two methods have demonstrated to be particularly effective in this context: coupling by reflection (Eberle, 2016; Eberle et al., 2019) and sticky coupling (Eberle & Zimmer, 2019). The flexibility and power of these methods are exemplified by recent developments in adapting these techniques to more complex dynamics such as McKean–Vlasov equations (Durmus et al., 2024; Cecchin et al., 2024). In particular, given the connection with stochastic control, significant assistance in analyzing the Wasserstein distance between controlled SDEs is provided by the so-called controlled coupling introduced in (Conforti, 2024).

**Related literature.** The power and appeal of SGMs have spurred significant interest in establishing convergence bounds, leading to a variety of mathematical frameworks.

Broadly, these contributions can be categorized into two primary approaches, focusing on different metrics and divergences.

The *first category* explores convergence bounds based on $\alpha$-divergences and Total Variation (TV) distance (see, *e.g.*, Block et al., 2020; De Bortoli, 2022b; De Bortoli et al., 2021b; Lee et al., 2022; 2023; Chen et al., 2023; 2022a; Oko et al., 2023). For example, Chen et al. (2022a) derived upper bounds in TV distance, assuming smoothness of the score function. Recent works by Conforti et al. (2023a) and Benton et al. (2024) extended these results to Kullback–Leibler (KL) divergence under milder assumptions about the data distribution. Importantly, bounds on KL divergence imply bounds on TV distance via Pinsker's inequality, reinforcing their broader applicability.

The *second category* focuses on convergence bounds in Wasserstein distances of order $p \geq 1$, which are often more practical for estimation tasks. De Bortoli (2022a) established $\mathcal{W}_1$ bounds with exponential rates under the manifold hypothesis, assuming the target distribution lies on a lower-dimensional manifold or represents an empirical distribution. Mimikos-Stamatopoulos et al. (2024) provided $\mathcal{W}_1$ bounds on a torus of radius $R$, noting that the bounds depend on $R$ and require early stopping criteria. For $\mathcal{W}_2$-convergence, results by De Bortoli et al. (2021a) and Lee et al. (2023) rely on smoothness assumptions about the score function or its estimator, often under bounded support conditions. For such distributions, $\mathcal{W}_2$-distance can also be bounded by TV distance, and thus by KL divergence, via Pinsker's inequality.

In addition to these general approaches, reverse SDEs have been analyzed in the context of log-concave sampling. For instance, Chen et al. (2022b) studied the proximal sampler algorithm introduced by Lee et al. (2021), inspiring further work on $\mathcal{W}_2$-convergence within the strongly log-concave framework. Current results often assume strong log-concavity of the data distribution and impose regularity conditions on the score function or its estimator (*e.g.*, Gao et al., 2023; Bruno et al., 2023; Tang & Zhao, 2024; Strasman et al., 2024). For strongly log-concave distributions, $\mathcal{W}_2$ bounds can also be estimated from KL divergence using Talagrand's inequality (Corollary 7.2, Gozlan & Léonard, 2010).

These works share some notable features. A key insight, as highlighted by Strasman et al. (2024), is the contraction property of the backward process, which ensures convergence stability. Additionally, their results align with well-established findings for Euler–Maruyama (EM) discretization schemes (Pagès, 2018), showing a $\sqrt{h}$ dependence on the step size $h$. Furthermore, Bruno et al. (2023) achieved optimal dimensional dependence in this framework, scaling as $\sqrt{d}$, where $d$ is the dimensionality of the data distribution.

**Contributions.** Our work breaks through the traditional constraints of log-concavity in data distributions and strict regularity requirements for the score function (or its estimator), offering a novel perspective on the $\mathcal{W}_2$ convergence of SGMs through coupling techniques.

Building on the concept of weak log-concavity (see, *e.g.*, Conforti, 2024), we establish a foundational framework for studying data distributions under significantly relaxed convexity assumptions. By leveraging the regularizing properties of the Ornstein–Uhlenbeck (OU) process, we demonstrate how its Gaussian stationary distribution progressively enhances the regularity of the initial data distribution. This regularization propagates weak log-concavity over time, ultimately transitioning into a strongly log-concave regime. Using a PDE-based approach inspired by Conforti et al. (2023a;b), we rigorously track the propagation of weak log-concavity through the HJB equation satisfied by the log-density of the forward process. Our analysis provides explicit estimates for the evolution of the weak log-concavity constant along the OU flow, culminating in a precise characterization of the transition to strong log-concavity (see Appendix B.3).

We also identify and analyze two regimes for the backward stochastic differential equation (SDE): a *contractive regime* and a *non-contractive one*. In Appendix B.3, we precisely quantify the transition point where the drift of the backward process ceases to be contractive, a critical insight for designing robust neural architectures and optimizing practical algorithms.

Additionally, we show that Gaussian mixtures inherently satisfy the weak log-concavity and log-Lipschitz assumptions required by our framework. In Proposition 4.1, we quantify these properties and demonstrate their compatibility with our approach. Moreover, our methods extend to the analysis of convolutions of densities with Gaussian kernels. The OU flow, which acts as a Gaussian kernel, regularizes the initial distribution, ensuring stability even under early stopping regimes. This versatility underscores the broad applicability of our results, transcending the assumptions specified in H1.

A key advantage of our approach is that it circumvents the strict regularity conditions on the score function and its estimator often imposed by prior studies (*e.g.*, Gao et al., 2023; Lee et al., 2022; Kwon et al., 2022; Bruno et al., 2023; Tang & Zhao, 2024). Instead, we rely solely on the mild assumptions of weak log-concavity and one-sided log-Lipschitz regularity of the data distribution. These assumptions suffice to derive the required score function regularity directly, eliminating the need for additional constraints (see Appendix B). This shift broadens the applicability of our framework while simplifying practical implementation.

Finally, we derive a fully explicit $\mathcal{W}_2$-convergence bound, with all constants explicitly dependent on the parameters of the data distribution. Unlike previous works, which often present these constants in non-explicit forms or as arbitrarily rescaled factors, our analysis provides transparency. This clarity enables precise assessment of how input parameters influence convergence, facilitating informed decisions when designing neural architectures and optimizing SGMs for real-world applications.

The structure of this paper is as follows. Section 2 introduces SGMs, presenting the general framework for analyzing these models and specifying the assumptions required to establish our convergence bounds. In Section 3, we focus on the main result: the convergence bound for SGMs in $\mathcal{W}_2$-distance, highlighting parallels with the key features identified in KL-divergence bounds for SGMs. Section 4 demonstrates the validity of our assumptions by showing that the general class of Gaussian mixtures satisfies the conditions necessary for the bound to hold, thereby establishing a strong connection with bounds in early stopping regimes. Section 5 explores the underlying features contributing to the remarkable success and effectiveness of SGMs, including the contractive properties of the OU flow, which form the basis for proving our result. Finally, Section 6 provides a concise sketch of the proof of the main result.

## 2. Score Generative Models

Let $\pi_{\text{data}} \in \mathcal{P}(\mathbb{R}^d)$ represent a probability distribution on $\mathbb{R}^d$, from which we want to generate samples. SGMs enable this by following a two-step approach: first, transforming data into noise, and second, learning how to reverse this process to recover data from noise.

To "create noise from data," SGMs employ an ergodic forward Markov process that begins with the data distribution and eventually converges to an invariant distribution, usually Gaussian. This invariant distribution serves as the starting "noise," which is easily generated by evolving the forward process from the data.

This corresponds to fixing a time horizon $T > 0$ and considering a $d$-dimensional ergodic diffusion over $[0, T]$ via the following SDE:

$$\mathrm{d}\overrightarrow{X}_t = \beta(\overrightarrow{X}_t)\mathrm{d}t + \Sigma \mathrm{d}B_t, \tag{1}$$

for $t \in [0, T]$, with $\beta : \mathbb{R}^d \to \mathbb{R}^d$ a drift function, $\Sigma \in \mathbb{R}^{d \times d}$ a fixed covariance matrix, and $(B_t)_{t \geq 0}$ a $d$-dimensional Brownian motion, initialized at $\overrightarrow{X}_0 \sim \pi_{\text{data}}$. Under mild assumptions on $\beta$, this equation admits unique solutions and is associated with a Markov semigroup $(P_t)_{t \geq 0}$ with a unique stationary distribution $\pi_\infty$. Moreover, the law of the process $\overrightarrow{X}_t$ admits a density $\overrightarrow{p}_t$ w.r.t. the Lebesgue measure.

To "create data from noise," the forward process is reversed using its time-reversal, known as the backward process. Specifically, we can sample from the noisy invariant distribution (which is easy to do), then apply the backward dynamics starting from these noisy samples. Since we are using the reversed process, at time $t = T$, the backward process ideally yields samples from the target distribution $\pi_{\text{data}}$.

More rigorously, SGMs aim at implementing the time-reversal process (see, *e.g.*, Anderson, 1982; Haussmann & Pardoux, 1986; Föllmer, 2005) defined by

$$\mathrm{d}\overleftarrow{X}_t = (-\beta(\overleftarrow{X}_t) + \Sigma\Sigma^\top \nabla \log \overrightarrow{p}_{T-t}(\overleftarrow{X}_t))\mathrm{d}t + \Sigma\mathrm{d}\bar{B}_t \,,$$

for $t \in [0, T]$, with $\overleftarrow{X}_0 \sim \mathcal{L}(\overrightarrow{X}_T)$ and $(\bar{B}_t)_{t\geq 0}$ a Brownian motion, explicitly characterized in Haussmann & Pardoux (1986, Remark 2.5).

To put into practice such procedure, three approximations must be made:

1. Since sampling from $\mathcal{L}(\overrightarrow{X}_T)$ is not feasible, the backward process is initialized at the easy-to-sample stationary distribution $\pi_\infty$.

2. The score function $(t, x) \mapsto \nabla \log \overrightarrow{p}_t(x)$ is unknown in closed form, as it depends on the (not directly accessible) distribution $\pi_{\text{data}}$, and thus needs to be estimated. This score function can be interpreted as a conditional expectation (see, *e.g.*, Equation 49, Conforti et al., 2023a), a key insight behind the success of SGMs. By leveraging the fact that a conditional expectation can be represented as an $L^2$-projection (see, *e.g.*, Corollary 8.17, Klenke, 2013), the score function can be estimated by training a model $\theta \mapsto s_\theta(t, x)$ to minimize the score-matching loss

$$\theta \mapsto \int_0^T \mathbb{E}\left[\|s_\theta(t, \overrightarrow{X}_t) - \nabla \log \overrightarrow{p}_t(\overrightarrow{X}_t)\|^2\right] \mathrm{d}t \,, \tag{2}$$

over a (rich enough) parametric family $\{s_\theta : \theta \in \Theta\}$.

3. As the continuous-time SDE can not be simulated exactly, the process is discretized, often using the Euler–Maruyama (EM) scheme or other stochastic integrators.

The resulting algorithm $(X_t^\star)_{t\in[0,T]}$ runs the EM scheme for the estimated backward process initialized at the stationary distribution of (1): for the learned parameter $\theta^\star$ and a sequence of step sizes $\{h_k\}_{k=1}^N$, $N \geq 1$, such that $\sum_{k=1}^N h_k = T$, we set $X_0^\star \sim \pi_\infty$ and compute for $k \in \{0, \ldots, N-1\}$

$$X_{t_{k+1}}^\star = X_{t_k}^\star + h_k\left(-\beta(X_{t_k}^\star) + s_{\theta^\star}(T - t_k, X_{t_k}^\star)\right)$$
$$+ \sqrt{2h_k}\Sigma Z_k \,,$$

with $\{Z_k\}_k$ a sequence of i.i.d. standard Gaussian random variables. Finally, this aims to return $\mathcal{L}(X_T^\star)$, an approximation of $\pi_{\text{data}}$.

## 2.1. OU-base Score Generative Models

We focus now on the OU case. In this case, $\pi_\infty$ is the standard Gaussian distribution, and forward and backward processes turn respectively into

$$\mathrm{d}\overrightarrow{X}_t = -\overrightarrow{X}_t\mathrm{d}t + \sqrt{2}\mathrm{d}B_t \,, \tag{3}$$
$$\mathrm{d}\overleftarrow{X}_t = (\overleftarrow{X}_t + 2\nabla\log\overrightarrow{p}_{T-t}(\overleftarrow{X}_t))\mathrm{d}t + \sqrt{2}\mathrm{d}B_t \,, \tag{4}$$

for $t \in [0, T]$. Since $\nabla\log\pi_\infty(x) = -x$, equation (4) can be reformulated equivalently as

$$\mathrm{d}\overleftarrow{X}_t = b_{T-t}(\overleftarrow{X}_t)\mathrm{d}t + \sqrt{2}\mathrm{d}B_t \,, \tag{5}$$

for $t \in [0, T]$, with, for $(t, x) \in [0, T] \times \mathbb{R}^d$,

$$b_t(x) := -x + 2\nabla\log\tilde{p}_t(x) \,, \tag{6}$$
$$\tilde{p}_t(x) := \overrightarrow{p}_t(x)/\pi_\infty(x) \,. \tag{7}$$

This framework, opposed to (4), has been adopted in several works (see, *e.g.*, Conforti et al., 2023a; Strasman et al., 2024), due to its advantage of maintaining the same sign for the drift term as in the forward dynamics.

In this article, we shall consider SGMs that generate approximate trajectories of the backward process based on its representation (5). This means that, for the learned parameter $\theta^\star$ and a sequence of step sizes $\{h_k\}_{k=1}^N$, with $N \geq 1$ such that

$$\sum_{k=1}^N h_k = T \,,$$

we consider the OU-based SGM described by $X_0^\star \sim \pi_\infty$ and

$$X_{t_{k+1}}^\star = X_{t_k}^\star + h_k\left(-X_{t_k}^\star + 2\tilde{s}_{\theta^\star}(T - t_k, X_{t_k}^\star)\right) + \sqrt{2h_k}Z_k \,, \tag{8}$$

for $k \in \{0, \ldots, N-1\}$, with $\theta^\star$ the minimizer of

$$\theta \mapsto \int_0^T \mathbb{E}\left[\|\tilde{s}_\theta(t, \overrightarrow{X}_t) - \nabla\log\tilde{p}_t(\overrightarrow{X}_t)\|^2\right]\mathrm{d}t \,, \tag{9}$$

over a properly chosen parametric class $\{\tilde{s}_\theta : \theta \in \Theta\}$.

## 3. Convergence guarantees for OU-Based SGMs

To understand the performance of the proposed SGM algorithm, we aim to provide quantitative error estimates

between the distribution $\mathcal{L}(X_T^\star)$ and the data distribution $\pi_{\text{data}}$.

First, for a given differentiable vector field $\beta$, define its weak convexity profile as

$$\kappa_\beta(r) = \inf_{x,y \in \mathbb{R}^d : \|x-y\|=r} \left\{ \frac{(\nabla\beta(x) - \nabla\beta(y))^\top (x-y)}{\|x-y\|^2} \right\}. \tag{10}$$

This function can be regarded as an integrated convexity lower bound for $\beta$, for points that are at distance $r > 0$. This definition frequently arises in applications of coupling methods to study the long-term behavior of Fokker–Planck equations (see, *e.g.*, Conforti, 2023; 2024). While $\kappa_\beta \geq 0$ directly corresponds to the convexity of $\beta$, using non-uniform lower bounds on $\kappa_\beta$ allows for the development of a more general notion of convexity, often called *weak convexity*.

**Definition 3.1.** We say that a vector field $\beta$ is weakly convex if its weak convexity profile $\kappa_\beta$ defined in (10) satisfies

$$\kappa_\beta(r) \geq \alpha - \frac{1}{r} f_M(r), \tag{11}$$

for some positive constants $\alpha, M > 0$, with $f_M$ defined as

$$f_M(r) := 2\sqrt{M} \tanh\left(r\sqrt{M}/2\right). \tag{12}$$

Moreover, we say that $\beta$ is weakly concave if $-\beta$ is weakly convex.

We are now able to state our assumptions.

**H1** The data distribution $\pi_{\text{data}}$ is absolutely continuous w.r.t. Lebesgue measure with $\pi_{\text{data}}(\mathrm{d}x) = \exp(-U(x))\,\mathrm{d}x$, for some function $U : \mathbb{R}^d \to \mathbb{R}$, such that

(i) $\nabla U$ is $L_U$-one-sided Lipschitz, with $L_U \geq 0$, *i.e.*,

$$(\nabla U(x) - \nabla U(y))^\top (x-y) \leq L_U \|x-y\|^2; \tag{13}$$

for any $x, y \in \mathbb{R}^d$;

(ii) $U$ is weakly convex, with weak convexity profile $\kappa_U$ satisfying

$$\kappa_U(r) \geq \alpha - \frac{1}{r} f_M(r), \tag{14}$$

for some $\alpha, M > 0$.

*Remark* 3.2. These weak assumptions represent a novel contribution to the literature on $\mathcal{W}_2$ convergence of SGMs. In fact, the entire class of Gaussian mixtures satisfies both

the weak log-concavity and (one-sided) log-Lipschitz continuity conditions. In Proposition 4.1, we provide explicit values for the weak log-concavity constants as well as the (one-sided) log-Lipschitz constants, illustrating the broad applicability and generality of these assumptions.

*Remark* 3.3. We remark that, as shown in Appendix B.1, Assumption H1 implies that $\pi_{\text{data}}$ has finite second order moment. In the sequel, we denote by

$$\mathrm{m}_2 := \int_{\mathbb{R}^d} \|x\|^2 \pi_{\text{data}}(\mathrm{d}x). \tag{15}$$

**H2** There exists $\varepsilon \geq 0$ and $\theta^\star \in \Theta$ such that for any $k \in \{0, ..., N\}$

$$\left\| \nabla \log \tilde{p}_{T-t_k}\left(X_{t_k}^\star\right) - \tilde{s}_{\theta^\star}\left(T - t_k, X_{t_k}^\star\right) \right\|_{L^2} \leq \varepsilon,$$

where the $L^2$-norm is defined as $\|\cdot\|_{L^2} := \mathbb{E}[\|\cdot\|^2]^{1/2}$.

*Remark* 3.4. Assumptions of this type have already been considered in the literature (see, *e.g.*, Gao et al., 2023; Bruno et al., 2023; Strasman et al., 2024). We remark that, since we take the expectation over the EM algorithm $(X_{t_k}^\star)_{k=0}^N$, for which the density is known, Assumption H2 is not only theoretically well-founded but also practically verifiable. This makes it a robust and applicable framework in real-world settings. In addition, in the simple case $\pi_{\text{data}} = \mathcal{N}(\mu, \mathbb{I})$ for some unknown $\mu \in \mathbb{R}^d$, when approximating the score function $\nabla \log \tilde{p}_t(x)$ by means of the neural networks $\{e^{-t}\theta : \theta \in \mathbb{R}^d\}$, Assumption H2 holds true for the minimizer $\theta = \mu$ of (2) and any $\varepsilon > 0$. Under the additional assumption that $\tilde{s}_{\theta^\star}(T - t_k, \cdot)$ is uniformly Lipschitz in space, Assumption H2 becomes fully compatible with the standard estimation error assumptions commonly used in the literature (see, *e.g.* Chen et al., 2022a; Li et al., 2023; Conforti et al., 2023a). This aligns with Proposition B.4, which establishes that the score function $\nabla \log \tilde{p}_t$ is Lipschitz continuous in space, for $t \in [0, T]$.

### 3.1. Main result

Under the assumptions stated above, we now present our main result. For clarity, we provide an informal version here and defer the formal statement–with fully explicit constants–to Appendix C.

**Theorem 3.5** (Informal). *Suppose that Assumption H1 and H2 hold. Consider the discretization $\{t_k, 0 \leq k \leq N\}$ of $[0, T]$ of constant step size $h$ small enough. Then, there exists a constant $C = C(\alpha, M, L_U) > 0$, which depends exponentially on the parameters $\alpha$, $M$, and $L_U$, such that*

$$\mathcal{W}_2\left(\pi_{\text{data}}, \mathcal{L}(X_{t_N}^\star)\right)$$
$$\leq C\left(e^{-T}\mathcal{W}_2\left(\pi_{\text{data}}, \pi_\infty\right) + \varepsilon T + \sqrt{h}\sqrt{d}\sqrt{\mathrm{m}_2}\,T\right).$$

*Remark* 3.6. We highlight the following.

- This $\mathcal{W}_2$-convergence bound aligns with recent literature on these models, both by generalizing results previously established for log-concave distributions and by identifying the key features of these models as captured through KL divergence: the initialization error decreases exponentially with $T$ (Conforti et al., 2023a; Strasman et al., 2024; Benton et al., 2024); the score-estimation error is proportional to $\varepsilon T$ (Conforti et al., 2023a; Lee et al., 2022; 2023; Chen et al., 2022a); the discretization error depends on $\sqrt{h \, \mathrm{m}_2 \, d}$ with $h$ mesh of the time-grid (Pagès, 2018), $\mathrm{m}_2$ second order moment of $\pi_{\mathrm{data}}$ (Conforti et al., 2023a; Chen et al., 2023), and $d$ dimension of the space (Bruno et al., 2023; Strasman et al., 2024; Gao et al., 2023). These features are present up to a finite multiplicative constant that depend on the parameters $\alpha$, $M$, and $L_U$ characterizing the data distribution $\pi_{\mathrm{data}}$, and that is fully explicit (see Theorem C.1 for its explicit expression).

- For sake of simplicity, the theorem is presented for a uniform mesh of the interval $[0, T]$. Proceeding as in Strasman et al. (2024), the analysis can be extended to a non-uniform subdivision of $[0, T]$ by considering the forward and backward SDEs (3)-(5) as time-inhomogeneous.

- Following Conforti et al. (2023a), the bound can be reformulated to depend directly on $\varepsilon$ by refining Assumption H2. As the solution $X_t$ to (3) converges in law to $\pi_\infty$, the modified score function $(t, x) \mapsto \nabla \log \tilde{p}_t(x)$ approaches zero for large $t$. Accounting for this behavior allows to scale the required precision $\varepsilon$ as $1/T$.

- This $\mathcal{W}_2$-bound exhibits the same dependencies on $T, \varepsilon, d, h$ and $\mathrm{m}_2$ as the KL-bounds provided in (*e.g.*) Chen et al. (2023) and Conforti et al. (2023a). A key distinction, however, lies in the fact that our result is expressed in terms of the Wasserstein distance $\mathcal{W}_2 (\pi_{\mathrm{data}}, \pi_\infty)$ rather than the KL divergence $\mathrm{KL}(\pi_{\mathrm{data}}|\pi_\infty)$, making its estimation from samples more practical (see, *e.g.*, Strasman et al., 2024), and is directly derived with the use of coupling techniques.

## 4. Gaussian mixture example

We demonstrate that Gaussian mixtures naturally fulfill the weak log-concavity and (one-sided) log-Lipschitz conditions central to our framework. As the OU flow—serving as a Gaussian kernel—regularizes the initial distribution, it guarantees stability even with early stopping. Consequently, our approach extends to the analysis of convolutions between densities and Gaussian kernels. This adaptability highlights the broad relevance of our findings, exceeding the specific assumptions outlined in H1.

**Proposition 4.1.** *Let $p_n$ be a Gaussian mixture on $\mathbb{R}^d$ having density law*

$$p_n(x) := \sum_{i=1}^n \beta_i \frac{1}{(2\pi\sigma_i^2)^{d/2}} \exp\left(-\frac{|x - \mu_i|^2}{2\sigma_i^2}\right),$$

*for $x \in \mathbb{R}^d$, with $\sigma_i > 0$, $\mu_i \in \mathbb{R}^d$ and $\beta_i \in [0, 1]$, for $i \in \{1, \ldots, n\}$, such that $\sum_{i=1}^n \beta_i = 1$. Then, $-\log p_n$ is weakly convex and $\nabla \log p_n$ is Lipschitz.*

In Appendix A, Proposition A.1, we also quantify and express the related constants.

*Remark* 4.2. The result presented above can be directly generalized to the case where the covariance matrices of each mode of the Gaussian mixture are of full rank but not scalar. In this more general setting, the parameter $1/\alpha_{p_n}$ (respectively, $1/\beta_{p_n}$) corresponds to the maximum (respectively, minimum) eigenvalue of the covariance matrices associated with each mode. Under the full-rank assumption, these eigenvalues are strictly positive, ensuring that the framework remains applicable and the bounds derived continue to hold with appropriately adjusted parameters.

## 5. Regime Switching

By exploiting the regularizing effects of the forward process, the weak log-concavity of the data distribution transitions to full log-concavity over time. Additionally, the drift of the time-reversed OU process alternates between contractive and non-contractive phase, reflecting the evolving concavity dynamics. By rigorously tracing how weak log-concavity propagates through the HJB equation governing the log-density of the forward process, we derive an exact expression for the critical moment $\xi(\alpha, M)$ at which the marginals of the forward process become strongly log-concave, as given in (22). We also provide an explicit lower bound $T(\alpha, M, 0)$, defined in (24), for the time $T^\star$ beyond which the drift of the backward OU process loses contractivity.

**Log-Concavity.**

- $\overrightarrow{p}_t$ is only weakly log-concave for $t \in [0, \xi(\alpha, M)]$;

- $\overrightarrow{p}_t$ is log-concave for $t \in [\xi(\alpha, M), T]$.

**Contractivity properties of the time-reversal process.**

- $b_t(x)$ is not (necessarily) contractive, for $t \in [0, T(\alpha, M, 0)]$ (*i.e.*, it is non-contractive in $[0, T^\star]$, while contractive in the interval $[T^\star, T(\alpha, M, 0)]$);

- $b_t(x)$ is contractive, for $t \in [T(\alpha, M, 0), T]$.

Further details and the explicit formulas for $\xi(\alpha, M)$ and $T(\alpha, M, 0)$ can be found in Appendix B.3.

This regime shift is a key element in the effectiveness of the SGMs. SDEs with contractive flows exhibit advantageous properties related to efficiency guarantees (see, *e.g.*,

Dalalyan, 2017; Durmus & Moulines, 2017; Cheng et al., 2018; Dwivedi et al., 2019; Shen & Lee, 2019; Cao et al., 2020; Mou et al., 2021; Li et al., 2021) that we can exploit in the low-time regime.

## 6. Sketch of the proof of the main result

We now present a sketch of the proof of Theorem 3.5. Our analysis is based on the observation that, in the practical implementation of the algorithm defined in (8), three successive approximations introduce distinct sources of error: time-discretization, initialization, and score-approximation. To analyze these errors, we work with time-continuous interpolations of the following four processes:

- **Backward OU process.** The time-reversal $(\overleftarrow{X}_t)_{t \in [0,T]}$ of the OU process defined in (5).

- **EM–discretization scheme.** The EM–approximation $(X_{t_k}^N)_{k=0}^N$ of the backward process (5), started at $X_0^N \sim \mathcal{L}(\overrightarrow{X}_T)$

- **Initialization error.** The EM–approximation $(X_{t_k}^\infty)_{k=0}^N$ of the backward process (5), started at $X_0^\infty \sim \pi_\infty$

- **Score approximation.** The generative process $(X_{t_k}^\star)_{k=0}^N$ defined in (8).

These auxiliary processes allow us to separately track the three sources of error. Using the triangle inequality, we obtain:

$$
\begin{aligned}
\mathcal{W}_2\Big(\pi_{\text{data}}, \mathcal{L}(X_{t_N}^\star)\Big) \le \; & \mathcal{W}_2\Big(\mathcal{L}(\overleftarrow{X}_T), \mathcal{L}(X_{t_N}^N)\Big) \\
& + \mathcal{W}_2\Big(\mathcal{L}(X_{t_N}^N), \mathcal{L}(X_{t_N}^\infty)\Big) \\
& + \mathcal{W}_2\Big(\mathcal{L}(X_{t_N}^\infty), \mathcal{L}(X_{t_N}^\star)\Big).
\end{aligned}
$$

**Bound on $\mathcal{W}_2\Big(\mathcal{L}(\overleftarrow{X}_T), \mathcal{L}(X_{t_N}^N)\Big)$.** Consider the synchronous coupling between $(\overleftarrow{X}_t)_{t \in [0,T]}$ and the continuous-time interpolation of $(X_{t_k}^N)_{k=0}^N$ with the same initialization, *i.e.* use the same Brownian motion to drive the two processes and set $\overleftarrow{X}_0 = X_0^N$. Then, it holds

$$
\mathcal{W}_2\Big(\mathcal{L}(\overleftarrow{X}_T), \mathcal{L}(X_{t_N}^N)\Big) \le \left\| \overleftarrow{X}_T - X_T^N \right\|_{L_2}.
$$

To bound the right-hand side, we aim to estimate $\|\overleftarrow{X}_{t_{k+1}} - X_{t_{k+1}}^N\|_{L_2}$ in terms of $\|\overleftarrow{X}_{t_k} - X_{t_k}^N\|_{L_2}$ and develop a recursion. Since we have considered the synchronous coupling

between $\overleftarrow{X}$ and $X^N$, we obtain

$$
\begin{aligned}
& \overleftarrow{X}_{t_{k+1}} - X_{t_{k+1}}^N \\
& = \overleftarrow{X}_{t_k} - X_{t_k}^N + \int_{t_k}^{t_{k+1}} \Big\{ -\Big(\overleftarrow{X}_t - X_{t_k}^N\Big) \\
& \quad + 2\Big(\nabla \log \tilde{p}_{T-t}\Big(\overleftarrow{X}_t\Big) - \nabla \log \tilde{p}_{T-t_k}\big(X_{t_k}^N\big)\Big) \Big\} \, \mathrm{d}t.
\end{aligned}
$$

By applying the triangle inequality, we obtain:

$$
\begin{aligned}
& \left\| \overleftarrow{X}_{t_{k+1}} - X_{t_{k+1}}^N \right\|_{L_2} \\
& \le \left\| \overleftarrow{X}_{t_k} - X_{t_k}^N \right. \\
& \quad + \int_{t_k}^{t_{k+1}} \mathrm{d}t \Big\{ -\Big(\overleftarrow{X}_{t_k} - X_{t_k}^N\Big) \\
& \quad + 2\Big(\nabla \log \tilde{p}_{T-t_k}\Big(\overleftarrow{X}_{t_k}\Big) - \nabla \log \tilde{p}_{T-t_k}\big(X_{t_k}^N\big)\Big) \Big\} \Big\|_{L_2} \\
& \quad + \left\| \int_{t_k}^{t_{k+1}} \mathrm{d}t \Big\{ -\Big(\overleftarrow{X}_t - \overleftarrow{X}_{t_k}\Big) \right. \\
& \quad \left. + 2\Big(\nabla \log \tilde{p}_{T-t}\Big(\overleftarrow{X}_t\Big) - \nabla \log \tilde{p}_{T-t_k}\Big(\overleftarrow{X}_{t_k}\Big)\Big) \Big\} \right\|_{L^2} \\
& =: A_{1,k} + A_{2,k},
\end{aligned}
$$

For the first term, we have:

$$
\begin{aligned}
& A_{1,k}^2 \\
& = \left\| \overleftarrow{X}_{t_k} - X_{t_k}^N \right\|_{L_2}^2 + h^2 \left\| -\Big(\overleftarrow{X}_{t_k} - X_{t_k}^N\Big) \right. \\
& \quad \left. + 2\Big(\nabla \log \tilde{p}_{T-t_k}\Big(\overleftarrow{X}_{t_k}\Big) - \nabla \log \tilde{p}_{T-t_k}\big(X_{t_k}^N\big)\Big) \right\|_{L_2}^2 \\
& \quad + 2h\mathbb{E}\left[ \Big(\overleftarrow{X}_{t_k} - X_{t_k}^N\Big)^\top \Big( -\Big(\overleftarrow{X}_{t_k} - X_{t_k}^N\Big) \right. \\
& \quad \left. + 2\Big(\nabla \log \tilde{p}_{T-t_k}\Big(\overleftarrow{X}_{t_k}\Big) - \nabla \log \tilde{p}_{T-t_k}\big(X_{t_k}^N\big)\Big)\Big) \right].
\end{aligned}
$$

To bound this term, we need regularity properties of the score function. One of the main challenges of this proof is to derive minimal assumptions on the data that ensure the minimal regularity properties of the score function required for the bound. To this aim, we adopt a PDE-based approach, drawing on the insights from Conforti et al. (2023a;b), noting that $(t,x) \mapsto -\log \tilde{p}_{T-t}(x)$ solves a HJB equation. Combining this with the regularizing effects of the OU process, we show in Appendix B that the regularity Assumptions H1 propagates naturally along the HJB equation:

- $(t,x) \mapsto \nabla \log \tilde{p}_{T-t}(x)$ turns out to be $L_{T-t}$-Lipschitz in space, with $L_{T-t}$ as in (27) and bounded by $L$ as in (27);

- $(t,x) \mapsto -\log \tilde{p}_{T-t}(x)$ remains weakly convex with

$$
\kappa_{-\log \tilde{p}_{T-t}}(r) \ge C_{T-t},
$$

and $C_{T-t}$ as in (20).

These regularity properties enable us to bound $A_{1,k}$ as:

$$A_{1,k} \leq \delta_k \left\| \overleftarrow{X}_{t_k} - X_{t_k}^N \right\|_{L_2},$$

for some $\delta_k$ depending on $L_{t_k}$ and $C_{t_k}$.

For the second term, we have

$$A_{2,k}^2 = \left\| \int_{t_k}^{t_{k+1}} \left\{ b_t(\overleftarrow{X}_t) - b_{t_k}(\overleftarrow{X}_{t_k}) \right\} \mathrm{d}t \right\|_{L^2}^2.$$

To bound this term, inspired by Conforti et al. (2023a, Proposition 2), we adopt a stochastic control perspective. We interpret the backward process (5) as the solution to a stochastic control problem and the term $(2\nabla \log \tilde{p}_{T-t}(\overleftarrow{X}_t))_{t \in [0,T]}$ as the solution to the adjoint equation within a stochastic maximum principle. This allows us to bound (up to a constant) $A_{2,k}$ with $h\sqrt{h}\sqrt{d}\sqrt{\mathrm{m}_2}$.

Combining the bounds on $A_{1,k}$ and $A_{2,k}$, we derive the recursion

$$\left\| \overleftarrow{X}_T - X_T^N \right\|_{L_2}$$

$$\leq \left\| \overleftarrow{X}_0 - X_0^N \right\|_{L_2} \prod_{\ell=0}^{N-1} \delta_\ell + Ch\sqrt{h}\sqrt{d}\sqrt{\mathrm{m}_2} \sum_{k=0}^{N-1} \prod_{\ell=k}^{N-1} \delta_\ell$$

$$= Ch\sqrt{h}\sqrt{d}\sqrt{\mathrm{m}_2} \sum_{k=0}^{N-1} \prod_{\ell=k}^{N-1} \delta_\ell.$$

Given the regime switching detailed in Section 5, we then analyze the low-time and large-time regimes separately, utilizing the contractive properties of SDEs at small times to establish $\delta_k \leq 1$, and applying brute force estimates for large times.

These remarks yields that

$$\sum_{k=0}^{N-1} \prod_{\ell=k}^{N-1} \delta_\ell \leq \frac{C}{h}T.$$

This let us conclude that

$$\mathcal{W}_2\left( \mathcal{L}(\overleftarrow{X}_T), \mathcal{L}(X_{t_N}^N) \right) \lesssim \sqrt{h}\sqrt{d}\sqrt{\mathrm{m}_2}\, T.$$

**Bound on** $\mathcal{W}_2\left( \mathcal{L}(X_{t_N}^N), \mathcal{L}(X_{t_N}^\infty) \right)$**.** Consider the synchronous coupling between the continuous-time interpolations of $(X_{t_k}^N)_{k=0}^N$ and $(X_{t_k}^\infty)_{k=0}^N$ with initialization satisfying $\mathcal{W}_2(\pi_\infty, \mathcal{L}(\overrightarrow{X}_T)) = \|X_0^\infty - X_0^N\|_{L^2}$. Then, it holds

$$\mathcal{W}_2\left( \mathcal{L}(X_{t_N}^N), \mathcal{L}(X_{t_N}^\infty) \right) \leq \left\| X_T^N - X_T^\infty \right\|_{L_2}.$$

By mirroring the previous argument and developing the recursion over the time intervals $[t_{k+1}, t_k]$, we get

$$\left\| X_T^N - X_T^\infty \right\|_{L_2} \leq \left\| X_0^N - X_0^\infty \right\|_{L_2} \prod_{\ell=0}^{N-1} \delta_\ell,$$

with the $\delta_k$s as before. We bound $\|X_0^\infty - X_0^N\|_{L^2}$ in the following (by now) standard way (see, *e.g.*, the proof of Proposition C.2, Strasman et al., 2024)

$$\left\| X_0^N - X_0^\infty \right\|_{L_2} = \mathcal{W}_2(\pi_\infty, \mathcal{L}(\overrightarrow{X}_T)) \leq \mathrm{e}^{-T}\mathcal{W}_2(\pi_{\text{data}}, \pi_\infty).$$

Using the previous considerations on the contractivity properties of the forward flow, we get that the product $\prod_{\ell=0}^{N-1} \delta_\ell$ is uniformly bounded by a constant depending on the parameters of the model. This yields to have

$$\mathcal{W}_2\left( \mathcal{L}(X_{t_N}^N), \mathcal{L}(X_{t_N}^\infty) \right) \lesssim \mathrm{e}^{-T}\mathcal{W}_2(\pi_{\text{data}}, \pi_\infty).$$

**Bound on** $\mathcal{W}_2\left( \mathcal{L}(X_{t_N}^\infty), \mathcal{L}(X_{t_N}^\star) \right)$**.** Consider the synchronous coupling between the continuous-time interpolations of $(X_{t_k}^\infty)_{k=0}^N$ and $(X_{t_k}^\star)_{k=0}^N$, with the same initialization, *i.e.*, $X_0^\infty = X_0^\star$. Using the evolution of these processes, together with the triangle inequality, we get

$$\left\| X_{t_{k+1}}^\infty - X_{t_{k+1}}^\star \right\|_{L_2}$$

$$\leq \left\| X_{t_k}^\infty - X_{t_k}^\star \right.$$

$$+ \int_{t_k}^{t_{k+1}} \mathrm{d}t \left\{ -(X_t^\infty - X_t^\star) \right.$$

$$\left. + 2\left( \nabla \log \tilde{p}_{T-t_k}\left(X_{t_k}^\infty\right) - \nabla \log \tilde{p}_{T-t_k}\left(X_{t_k}^\star\right) \right) \right\} \right\|_{L_2}$$

$$+ 4 \left\| \int_{t_k}^{t_{k+1}} \mathrm{d}t \right.$$

$$\left. \left( \nabla \log \tilde{p}_{T-t_k}\left(X_{t_k}^\star\right) - \tilde{s}_{\theta^\star}\left(T - t_k, X_{t_k}^\star\right) \right) \right\|_{L^2}$$

$$=: B_{1,k} + B_{2,k}.$$

By reproposing a similar argument as before, we get

$$B_{1,k} \leq \delta_k \left\| X_{t_k}^\infty - X_{t_k}^\star \right\|_{L_2}.$$

By using Assumption H2, we obtain $B_{2,k} \leq 4h\epsilon$. Therefore, developing the recursion as before, we derive the bound

$$\mathcal{W}_2\left( \mathcal{L}(X_{t_N}^\infty), \mathcal{L}(X_{t_N}^\star), \right) \lesssim \varepsilon T.$$

## 7. Conclusions

This paper presents a unified framework for deriving $\mathcal{W}_2$-convergence bounds for SGMs, leveraging both PDE and stochastic control approaches, while relaxing strong regularity assumptions on the data distribution, the score function and its estimator. The results mark a significant advancement, requiring only weak log-concavity and one-sided log-Lipschitz conditions on the data distribution, with no regularity needed for the score or its estimator. This broadens applicability to diverse data types.

Using Gaussian mixtures, we illustrate the versatility of our framework and show how Gaussian kernel convolutions improve early-stopping methods via regularization. We also analyze how weak log-concavity evolves into full log-concavity over time, and how the drift of the time-reversed OU process shifts between contractive and non-contractive regimes, reflecting this transition.

While our bound depends exponentially on $\alpha$, $M$, and $L_U$, it would be valuable to investigate whether this can be improved to a polynomial dependence—a question we leave for future work.

## Acknowledgements

We gratefully acknowledge Giovanni Conforti, Alain Durmus, and Gabriel V. Cardoso for their insightful discussions and valuable feedback throughout the development of this work. We also thank the anonymous reviewers for their constructive comments, which significantly improved the quality of this paper.

The work of M.G.S. has been supported by the Paris Ile-de-France Région in the framework of DIM AI4IDF. The work of A.O. was funded by the European Union (ERC-2022-SYG-OCEAN-101071601). Views and opinions expressed are however those of the author only and do not necessarily reflect those of the European Union or the European Research Council Executive Agency. Neither the European Union nor the granting authority can be held responsible for them.

## Impact Statement

This paper presents work whose goal is to advance the field of Machine Learning. There are many potential societal consequences of our work, none which we feel must be specifically highlighted here.

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

## Appendix

The appendix includes the additional materials to support the findings and analyses presented in the main paper. **Appendix A** delves into the Gaussian mixture example, establishing its weak log-concavity and log-Lipschitz properties with explicit derivations of the associated constants. **Appendix B** focuses on the propagation of regularity assumptions through the forward OU process, leveraging PDE-based techniques to demonstrate how weak log-concavity and Lipschitz regularity evolve over time. In this section, we also examine the dynamics of concavity and contractivity in the OU process, rigorously characterizing transitions from weak to strong log-concavity and identifying contractive and non-contractive regimes. **Appendix C** provides the formal counterpart of Theorem 3.5 and the proof of it. Finally, **Appendix D** compiles auxiliary results and technical lemmas needed to develop the argument carried on in the proof of the main result.

## A. Gaussian mixture example

**Proposition A.1.** *Let $p_n$ be a Gaussian mixture in $\mathbb{R}^d$ having density law*

$$p_n := \sum_{i=1}^{n} \beta_i \frac{1}{(2\pi\sigma_i^2)^{d/2}} \exp\left(-\frac{|x-\mu_i|^2}{2\sigma_i^2}\right), \quad x \in \mathbb{R}^d, \tag{16}$$

*with $\sigma_i > 0$, $\mu_i \in \mathbb{R}^d$ and $\beta_i \in [0,1]$, for $i \in \{1,\dots,n\}$, such that $\sum_{i=1}^{n} \beta_i = 1$. Then, $-\log p_n$ is weakly convex with coefficients*

$$\alpha_{p_n} = \frac{1}{\max_{i \in \{0,\dots,n\}} \sigma_i^2}, \qquad \sqrt{M_{p_n}} := 2n \sum_{i=1}^{n} \frac{\|\mu_i\|}{\sigma_i^2}.$$

*Moreover, we have that $\nabla \log p_n$ is $(\beta_{p_n} + \sqrt{M_{p_n}})$-Lipschitz, with*

$$\beta_{p_n} = \frac{1}{\min_{i \in \{0,\dots,n\}} \sigma_i^2}.$$

*Proof of Proposition 4.1.* **Step 1. Gaussian mixture of two equi-weighted modes.** Consider, first, the following Gaussian mixture of two modes with equal weight, *i.e.*, $\beta_1 = \beta_2 = 1/2$, each mode having same variance $\sigma^2 \mathbb{I}$. Remark that the property of being weakly log-concave is invariant to translation. Therefore, up to a translation, we have that the density distribution of this law is

$$p_2(x) = \frac{1}{2} \frac{1}{(2\pi\sigma^2)^{d/2}} \exp\left(-\frac{|x-\mu|^2}{2\sigma^2}\right) + \frac{1}{2} \frac{1}{(2\pi\sigma^2)^{d/2}} \exp\left(-\frac{|x+\mu|^2}{2\sigma^2}\right),$$

for $x \in \mathbb{R}^d$, with $\mu \in \mathbb{R}^d$. This means that its score function is equal to

$$\nabla \log p_2(x) = -\frac{x}{\sigma^2} + \frac{\mu}{\sigma^2} \frac{\exp\left(-\frac{|x-\mu|^2}{2\sigma^2}\right) - \exp\left(-\frac{|x+\mu|^2}{2\sigma^2}\right)}{\exp\left(-\frac{|x-\mu|^2}{2\sigma^2}\right) + \exp\left(-\frac{|x+\mu|^2}{2\sigma^2}\right)} = -\frac{x}{\sigma^2} + \frac{\mu}{\sigma^2} \frac{\exp\left(\mu^\top x/\sigma^2\right) - 1}{\exp\left(\mu^\top x/\sigma^2\right) + 1}.$$

We focus now on bounding $\kappa_{-\log p}$. Fix $x, y \in \mathbb{R}^d$. We have that

$$-\frac{(\nabla \log p_2(x) - \nabla \log p_2(y))^\top (x-y)}{\|x-y\|^2} = \frac{1}{\sigma^2} - \frac{\mu^\top(x-y)}{\sigma^2 \|x-y\|^2} \times \left(\frac{\exp\left(\mu^\top x/\sigma^2\right) - 1}{\exp\left(\mu^\top x/\sigma^2\right) + 1} - \frac{\exp\left(\mu^\top y/\sigma^2\right) - 1}{\exp\left(\mu^\top y/\sigma^2\right) + 1}\right)$$

$$= \frac{1}{\sigma^2} - \frac{\mu^\top(x-y)}{\sigma^2 \|x-y\|^2} \times \left(\frac{2\left(\exp\left(\mu^\top x/\sigma^2\right) - \exp\left(\mu^\top y/\sigma^2\right)\right)}{\left(\exp\left(\mu^\top x/\sigma^2\right) + 1\right)\left(\exp\left(\mu^\top y/\sigma^2\right) + 1\right)}\right)$$

$$= \frac{1}{\sigma^2} - \frac{\mu^\top(x-y)}{\sigma^2 \|x-y\|^2} \times \left(\frac{2\left(\exp\left(\mu^\top (x-y)/\sigma^2\right) - 1\right)}{\left(\exp\left(\mu^\top x/\sigma^2\right) + 1\right)\left(\exp\left(-\mu^\top y/\sigma^2\right) + 1\right)}\right)$$

$$\geq \frac{1}{\sigma^2} - \frac{\mu^\top(x-y)}{\sigma^2 \|x-y\|^2} \times \left(\frac{2\left(\exp\left(\mu^\top (x-y)/\sigma^2\right) - 1\right)}{\left(\exp\left(\mu^\top (x-y)/\sigma^2\right) + 1\right)}\right)$$

$$= \frac{1}{\sigma^2} - 2\frac{\mu^\top(x-y)}{\sigma^2 \|x-y\|^2} \times \tanh\left(\frac{\mu^\top (x-y)}{2\sigma^2}\right).$$

It is clear that, the minimum of the r.h.s. of the previous inequality, under the constraint $|x - y| = r$, is reached for a vector $x - y$ that is colinear with $\mu$. Taking $x = y - \frac{\mu}{|\mu|} r$, we get

$$\kappa_{-\log p_2}(r) \geq \frac{1}{\sigma^2} - 2\frac{\|\mu\|}{\sigma^2} r^{-1} \tanh\left(\frac{\|\mu\|}{2\sigma^2} r\right).$$

From the definition of $f_M$ as in (12) and the fact that the function $M \mapsto f_M(r)$ is increasing for a fixed $r > 0$, we can take

$$\alpha_{p_2} := \frac{1}{\sigma^2}, \qquad \sqrt{M_{p_2}} := 4\sqrt{2}\frac{\|\mu\|}{\sigma^2}.$$

Moreover, recalling the expression we got for $\nabla \log p_2$ and repeating similar computations, we obtain that

$$\|\nabla \log p_n(x) - \nabla \log p_n(y)\| \leq \frac{1}{\sigma^2} \|x - y\| + \left\|\tanh\left(\frac{\mu^\top (x - y)}{2\sigma^2}\right)\right\| \left\|\frac{\mu}{\sigma^2}\right\|$$

$$\leq \frac{1}{\sigma^2} \|x - y\| + \left(\frac{\|\mu\|}{\sigma^2}\right)^2 \|x - y\| \leq (\beta_{p_2} + M_{p_2}) \|x - y\|,$$

where, in the second inequality we have used the sub-linearity of $x \mapsto \tanh x$ together with Cauchy-Schwartz inequality.

**Step 2. General Gaussian mixture.** Consider $p_n$ defined as in (16). Therefore, its score function is

$$\nabla \log p_n(x) = \frac{1}{p_n(x)} \sum_{i=1}^{n} \left[-\beta_i \frac{x - \mu_i}{\sigma_i^2} \times \frac{1}{(2\pi\sigma_i^2)^{d/2}} \exp\left(-\frac{\|x - \mu_i\|^2}{2\sigma_i^2}\right)\right]$$

$$= -\frac{x}{p_n(x)} \sum_{i=1}^{n} \frac{\beta_i}{\sigma_i^2} \frac{1}{(2\pi\sigma_i^2)^{d/2}} \exp\left(-\frac{\|x - \mu_i\|^2}{2\sigma_i^2}\right) + \frac{1}{p_n(x)} \sum_{i=1}^{n} \frac{\beta_i}{\sigma_i^2} \mu_i \frac{1}{(2\pi\sigma_i^2)^{d/2}} \exp\left(-\frac{\|x - \mu_i\|^2}{2\sigma_i^2}\right).$$

Fix $r > 0$ and $x, y \in \mathbb{R}^d$ such that $\|x - y\| = r$. We then have

$$-\frac{1}{r^2}\left(\nabla \log p_n(x) - \nabla \log p_n(y)\right)^\top (x - y)$$

$$= \frac{1}{r^2}\left\{\frac{x}{p_n(x)} \sum_{i=1}^{n} \frac{\beta_i}{\sigma_i^2} \frac{1}{(2\pi\sigma_i^2)^{d/2}} \exp\left(-\frac{\|x - \mu_i\|^2}{2\sigma_i^2}\right) - \frac{1}{p_n(x)} \sum_{i=1}^{n} \frac{\beta_i}{\sigma_i^2} \mu_i \frac{1}{(2\pi\sigma_i^2)^{d/2}} \exp\left(-\frac{\|x - \mu_i\|^2}{2\sigma_i^2}\right)\right.$$

$$\left. - \frac{y}{p_n(y)} \sum_{i=1}^{n} \frac{\beta_i}{\sigma_i^2} \frac{1}{(2\pi\sigma_i^2)^{d/2}} \exp\left(-\frac{\|y - \mu_i\|^2}{2\sigma_i^2}\right) + \frac{1}{p_n(y)} \sum_{i=1}^{n} \frac{\beta_i}{\sigma_i^2} \mu_i \frac{1}{(2\pi\sigma_i^2)^{d/2}} \exp\left(-\frac{\|y - \mu_i\|^2}{2\sigma_i^2}\right)\right\}^\top (x - y)$$

$$= \frac{1}{r^2}\left[\frac{x}{p_n(x)} \sum_{i=1}^{n} \frac{\beta_i}{\sigma_i^2} \frac{1}{(2\pi\sigma_i^2)^{d/2}} \exp\left(-\frac{\|x - \mu_i\|^2}{2\sigma_i^2}\right) - \frac{y}{p_n(y)} \sum_{i=1}^{n} \frac{\beta_i}{\sigma_i^2} \frac{1}{(2\pi\sigma_i^2)^{d/2}} \exp\left(-\frac{\|y - \mu_i\|^2}{2\sigma_i^2}\right)\right]^\top (x - y)$$

$$- \frac{1}{r^2}\left\{\frac{1}{p_n(x)} \sum_{i=1}^{n} \frac{\beta_i}{\sigma_i^2} \mu_i \frac{1}{(2\pi\sigma_i^2)^{d/2}} \exp\left(-\frac{\|x - \mu_i\|^2}{2\sigma_i^2}\right)\right.$$

$$\left. - \frac{1}{p_n(y)} \sum_{i=1}^{n} \frac{\beta_i}{\sigma_i^2} \mu_i \frac{1}{(2\pi\sigma_i^2)^{d/2}} \exp\left(-\frac{\|y - \mu_i\|^2}{2\sigma_i^2}\right)\right\}^\top (x - y)$$

$$= A_1 + A_2.$$

We now proceed in bounding the two terms in the r.h.s. of the previous equation. We see that

$$A_1 = \frac{1}{r^2}\left[x\,\psi_n(x) - y\,\psi_n(y)\right]^\top (x - y),$$

with

$$\psi_n(x) := \frac{1}{p_n(x)} \sum_{i=1}^{n} \frac{\beta_i}{\sigma_i^2} \frac{1}{(2\pi\sigma_i^2)^{d/2}} \exp\left(-\frac{\|x - \mu_i\|^2}{2\sigma_i^2}\right).$$

Without loss of generality, since $x$ and $y$ are interchangeable, we can suppose that $\psi_n(y) \le \psi_n(x)$. From the definition of $p_n$ as in (16), we have that $\alpha_{p_n} \le \psi_n(x)$ and, therefore,

$$A_1 = \frac{1}{r^2} \left[ x\, \psi_n(x) - y\, \psi_n(y) \right]^\top (x - y) \ge \frac{1}{r^2} \psi_n(x)(x-y)^\top (x-y) = \psi_n(x) \ge \alpha_{p_n}\,.$$

Consider now $A_2$ and denote $\mathfrak{e}_i(z) := \frac{1}{(2\pi\sigma_i^2)^{d/2}} \exp\left(-\frac{\|z-\mu_i\|^2}{2\sigma_i^2}\right)$, for $z \in \mathbb{R}^d$ and $i \in \{1,\dots,n\}$. Using a telescopic sum, we have

$$
\begin{aligned}
A_2 &= -\sum_{i=1}^n \frac{1}{r^2} \left[ \frac{\sum_{j=1}^{i-1} \beta_j \frac{\mu_j}{\sigma_j^2} \mathfrak{e}_j(x) + \beta_i \frac{\mu_i}{\sigma_i^2} \mathfrak{e}_i(x) + \sum_{j=i+1}^n \beta_j \frac{\mu_j}{\sigma_j^2} \mathfrak{e}_j(y)}{\sum_{j=1}^{i-1} \beta_j \mathfrak{e}_j(x) + \beta_i \mathfrak{e}_i(x) + \sum_{j=i+1}^n \beta_j \mathfrak{e}_j(y)} \right. \\
&\qquad\qquad \left. - \frac{\sum_{j=1}^{i-1} \beta_j \frac{\mu_j}{\sigma_j^2} \mathfrak{e}_j(x) + \beta_i \frac{\mu_i}{\sigma_i^2} \mathfrak{e}_i(y) + \sum_{j=i+1}^n \beta_j \frac{\mu_j}{\sigma_j^2} \mathfrak{e}_j(y)}{\sum_{j=1}^{i-1} \beta_j \mathfrak{e}_j(x) + \beta_i \mathfrak{e}_i(y) + \sum_{j=i+1}^n \beta_j \mathfrak{e}_j(y)} \right]^\top (x-y) \\
&= -\sum_{i=1}^n \frac{1}{r^2} \left[ \frac{v_i + \beta_i \frac{\mu_i}{\sigma_i^2} \mathfrak{e}_i(x)}{\gamma_i + \beta_i \mathfrak{e}_i(x)} - \frac{v_i + \beta_i \frac{\mu_i}{\sigma_i^2} \mathfrak{e}_i(y)}{\gamma_i + \beta_i \mathfrak{e}_i(y)} \right]^\top (x-y)\,,
\end{aligned}
$$

with

$$v_i := \sum_{j=1}^{i-1} \beta_j \frac{\mu_j}{\sigma_j^2} \mathfrak{e}_j(x) + \sum_{j=i+1}^n \beta_j \frac{\mu_j}{\sigma_j^2} \mathfrak{e}_j(y) \quad \text{and} \quad \gamma_i := \sum_{j=1}^{i-1} \beta_j \mathfrak{e}_j(x) + \sum_{j=i+1}^n \beta_j \mathfrak{e}_j(y)\,.$$

This means that, using that

$$(\gamma_i + \beta_i \mathfrak{e}_i(x))(\gamma_i + \beta_i \mathfrak{e}_i(y)) \ge \gamma_i \beta_i \mathfrak{e}_i(x) + \gamma_i \beta_i \mathfrak{e}_i(y)$$

since all $\gamma_i^2 \ge 0$, we get

$$
\begin{aligned}
A_2 &= -\sum_{i=1}^n \frac{1}{r^2} \left[ \frac{\beta_i(\mathfrak{e}_i(x) - \mathfrak{e}_i(y))\left(\gamma_i \frac{\mu_i}{\sigma_i^2} - v_i\right)}{(\gamma_i + \beta_i \mathfrak{e}_i(x))(\gamma_i + \beta_i \mathfrak{e}_i(y))} \right]^\top (x-y) \\
&\ge -\sum_{i=1}^n \frac{1}{r^2} \left[ \frac{\beta_i(\mathfrak{e}_i(x) - \mathfrak{e}_i(y))\left(\gamma_i \frac{\mu_i}{\sigma_i^2} - v_i\right)}{\beta_i \gamma_i (\mathfrak{e}_i(x) + \mathfrak{e}_i(y))} \right]^\top (x-y) \\
&= -\sum_{i=1}^n \frac{1}{r^2} \frac{\mathfrak{e}_i(x) - \mathfrak{e}_i(y)}{\mathfrak{e}_i(x) + \mathfrak{e}_i(y)} \left( \frac{\mu_i}{\sigma_i^2} - \frac{v_i}{\gamma_i} \right)^\top (x-y) \\
&= -\sum_{i=1}^n \frac{1}{r^2} \tanh\left( \frac{\mu_i^\top (x-y)}{2\sigma_i^2} \right) \left( \frac{\mu_i}{\sigma_i^2} \right)^\top (x-y) + \sum_{i=1}^n \frac{1}{r^2} \tanh\left( \frac{\mu_i^\top (x-y)}{2\sigma_i^2} \right) \left( \frac{v_i}{\gamma_i} \right)^\top (x-y)\,.
\end{aligned}
$$

First, note that, as in the *Step 1*, we have that

$$\frac{1}{r^2} \tanh\left( \frac{\mu_i^\top (x-y)}{\sigma_i^2} \right) \left( \frac{\mu_i}{\sigma_i^2} \right)^\top (x-y) \le \frac{1}{r} f_{M_i}(r)\,, \qquad \text{with } \sqrt{M_i} := \frac{\|\mu_i\|}{\sigma_i^2}\,.$$

Secondly, for a fixed $r > 0$, we see that the function $f_M(r)$ is increasing in $M$. Therefore, we get

$$-\sum_{i=1}^n \frac{1}{r^2} \tanh\left( \frac{\mu_i^\top (x-y)}{2\sigma_i^2} \right) \left( \frac{\mu_i}{\sigma_i^2} \right)^\top (x-y) \ge -n\frac{1}{r} f_{\hat{M}}(r)\,, \qquad \text{with } \sqrt{\hat{M}} := \sum_{i=1}^n \frac{\|\mu_i\|}{\sigma_i^2}\,.$$

Note that $v_i/\gamma_i$ is a convex combination of the vectors $\{\mu_j/\sigma_j^2\}_{j\neq i}$. Therefore, using again that $M \mapsto f_M(r)$ is increasing for a fixed $r > 0$, we have

$$\sum_{i=1}^n \frac{1}{r^2} \tanh\left(\frac{\mu_i^\top(x-y)}{2\sigma_i^2}\right) \left(\frac{v_i}{\gamma_i}\right)^\top (x-y) \geq -n\frac{1}{r}f_{\hat M}(r)\,.$$

Combining the previous bounds, together with monotonicity of $M \mapsto \tanh(\sqrt{M}r/2)$ for fixed $r > 0$, we can conclude that

$$A_2 \geq -\frac{1}{r}f_{M_{p_n}}(r)\,, \qquad \text{with } \sqrt{M_{p_n}} := 2n\sum_{i=1}^n \frac{\|\mu_i\|}{\sigma_i^2}\,.$$

Combining the bound on $A_1$ with the one on $A_2$, we can conclude that $-\log(p_n)$ is weakly convex with parameters $\alpha_{p_n}$ and $M_{p_n}$.

We now bound $\|\nabla \log p_n(x) - \nabla \log p_n(y)\|$. Following the same lines as before, we get

$$\|\nabla \log p_n(x) - \nabla \log p_n(y)\| \leq \|x\,\psi_n(x) - y\,\psi_n(y)\| + \sum_{i=1}^n \left|\tanh\left(\frac{\mu_i^\top(x-y)}{2\sigma_i^2}\right)\right| \left\|\frac{\mu_i}{\sigma_i^2} - \frac{v_i}{\gamma_i}\right\|$$

$$:= B_1 + B_2\,.$$

From the definition of $p_n$ as in (16), we have that $\psi_n(x) \leq \beta_{p_n}$ and, therefore,

$$B_1 \leq \max\{\psi_n(x), \psi_n(y)\}\,\|x-y\| \leq \beta_{p_n}\|x-y\|\,.$$

Analogously to the bound on $A_2$, we now use the sub-linearity of $x \mapsto \tanh x$, Cauchy-Schwartz inequality and the triangle one, together with the fact that $v_i/\gamma_i$ is a convex combination of the vectors $\{\mu_j/\sigma_j^2\}_{j\neq i}$, to get

$$B_2 \leq \sum_{i=1}^n \left|\frac{\mu_i^\top(x-y)}{2\sigma_i^2}\right| \left\|\frac{\mu_i}{\sigma_i^2} - \frac{v_i}{\gamma_i}\right\| \leq \|x-y\| \sum_{i=1}^n \left\|\frac{\mu_i}{\sigma_i^2}\right\| \left(\left\|\frac{\mu_i}{\sigma_i^2}\right\| + \left\|\frac{v_i}{\gamma_i}\right\|\right)$$

$$\leq 2n\sum_{i=1}^n \left(\frac{\|\mu_i\|}{\sigma_i^2}\right)^2 \|x-y\| \leq M_{p_n}\|x-y\|\,.$$

Putting together these inequalities, we obtain that $\nabla \log(p_n)$ is Lipschitz with Lipschitz constant $(\beta_{p_n} + M_{p_n})$. $\qquad\square$

## B. Propagation of the assumptions

The proof of our $\mathcal{W}_2$-convergence bound is based on some regularity properties of the score function $(t, x) \mapsto \nabla \log \tilde p_{T-t}(x)$. Therefore, in this section, we establish the key regularity properties of $\nabla \log \tilde p$, which arise from the propagation of Assumption H1 through the OU process flow (3).

### B.1. Weak-log concavity implies finite second order moment

**Proposition B.1.** *Suppose that Assumption H1(ii) holds. Then, $\pi_{\text{data}}$ admits a second order moment.*

*Proof.* Consider the following Taylor development up to order two.

$$U(x) = U(0) + \nabla U(x)^\top x + \frac{1}{2}x^\top \nabla^2 U(y)x\,,$$

for some $y \in \{tx\,:\,t \in [0,1]\}$. From Assumption H1(ii), we have that

$$-\nabla U(x)^\top x \leq -\alpha\|x\|^2 + f_M(\|x\|)\|x\| \leq -\alpha\|x\|^2 + M\|x\|\,.$$

This means that outside a ball $B(0, R)$ with $R$ big enough, there exists $\alpha_R > 0$ such that

$$-\nabla U(x)^\top x \leq -\alpha_R\|x\|^2\,, \qquad \text{for } x \notin B(0, R)\,.$$

From Bouchut et al. (2005, Lemma 2.2), we get that the previous inequality implies that

$$\nabla^2 U \preccurlyeq -\alpha_R I_d \,.$$

Therefore,

$$
\begin{aligned}
&\int_{\mathbb{R}^d} \|x\|^2 \exp\left(-U(x)\right) \mathrm{d}x \\
&= \int_{B(0,R)} \|x\|^2 \exp\left(-U(x)\right) \mathrm{d}x + \int_{\mathbb{R}^d \setminus B(0,R)} \|x\|^2 \exp\left(-U(x)\right) \mathrm{d}x \\
&\leq \int_{B(0,R)} \|x\|^2 \exp\left(-U(x)\right) \mathrm{d}x + \int_{\mathbb{R}^d \setminus B(0,R)} \|x\|^2 \exp\left(-U(0) - \frac{3}{2}\alpha_2\|x\|^2\right) < \infty \,.
\end{aligned}
$$

$\square$

### B.2. Weak-convexity of the modified score function

As highlighted in Conforti et al. (2023a), we observe that the function $(t,x) \mapsto -\log \tilde{p}_{T-t}(x)$ is a solution to a HJB equation. This observation allows us to establish an intriguing connection between the study of this class of non-linear PDEs and SGMs, as further explored in Conforti et al. (2023a); Gentiloni-Silveri et al. (2024).

First, we leverage the invariance of the class of weakly convex functions for the HJB equation satisfied by the log-density of the OU process (demonstrated by Conforti (2024)) to show how weak log-concavity propagates along the flow of (3). To this end, let $(S_t)_{t \geq 0}$ denote the semigroup generated by a standard Brownian motion on $\mathbb{R}^d$, $i.e.$,

$$S_t f(x) = \int \frac{1}{(2\pi t)^{d/2}} \exp\left(-\frac{\|x-y\|^2}{2t}\right) f(y)\mathrm{d}y \,, \tag{17}$$

with $f$ a general test function.

**Theorem B.2** (Theorem 2.1 in Conforti (2024)). *Consider the class*

$$\mathcal{F}_M := \left\{ g \in C^1(\mathbb{R}^d) \; : \; \kappa_g(r) \geq -f_M(r)r^{-1} \right\} \,.$$

*Then, we have that*

$$h \in \mathcal{F}_M \quad \Rightarrow \quad -\log S_t \mathrm{e}^{-h} \in \mathcal{F}_M \,, \qquad for\ t \geq 0 \,. \tag{18}$$

Theorem B.2 provides a substantial generalization of the the principle "*once log-concave, always log-concave*" by Saremi et al. (2023), to the weak log-concave setting. Indeed, when $M = 0$, $i.e.$, $\mathrm{e}^{-h}$ is log-concave, (18) implies that $S_t \mathrm{e}^{-h}$ remains log-concave. We remark that in the log-concave setting, (18) follows directly from the Prékopa-Leindler inequality, whose application is central to the findings in Bruno et al. (2023); Gao et al. (2023); Strasman et al. (2024).

We can now examine how the constant of weak log-concavity propagates. This result corresponds to Conforti et al. (2023b, Lemma 5.9), under the same set of assumptions.

**Lemma B.3** (Lemma 5.9 in Conforti et al. (2023b)). *Assume that Assumption H1(ii) holds and fix $t \in [0,T]$. Then, the function $x \mapsto -\log \tilde{p}_{T-t}(x)$ is weakly convex with weak convexity profile $\tilde{k}_t := \kappa_{-\log \tilde{p}_{T-t}}$ satisfying*

$$\tilde{k}_t(r) \geq \frac{\alpha}{\alpha + (1-\alpha)\mathrm{e}^{-2(T-t)}} - 1 - \frac{\mathrm{e}^{-(T-t)}}{\alpha + (1-\alpha)\mathrm{e}^{-2(T-t)}} \frac{1}{r} f_M\left(\frac{\mathrm{e}^{-(T-t)}}{\alpha + (1-\alpha)\mathrm{e}^{-2(T-t)}} r\right) \,, \tag{19}$$

*with $C_{T-t}$ given by*

$$C_t = \frac{\alpha}{\alpha + (1-\alpha)\mathrm{e}^{-2t}} - \frac{\mathrm{e}^{-2t}}{(\alpha + (1-\alpha)\mathrm{e}^{-2t})^2} M - 1 \,. \tag{20}$$

*In particular, the modified score function $x \mapsto \nabla \log \tilde{p}_{T-t}(x)$ satisfies*

$$\left(\nabla \log \tilde{p}_{T-t}(x) - \nabla \log \tilde{p}_{T-t}(y)\right)^\top (x-y) \leq -C_{T-t} \|x-y\|^2 \,, \qquad for\ x,y \in \mathbb{R}^d \,. \tag{21}$$

This lemma relies on the fact that the the flow of the OU process (3) can be rewritten w.r.t. the flow of the Brownian motion as

$$\tilde{p}_t(x) = S_{1-e^{-2t}} e^{-\left(U - \|\cdot\|^2/2\right)} \left(e^{-t}x\right) .$$

This remark enables the application of Theorem B.2, to obtain an estimation of the constant of weak log-concavity $C_t$.

## B.3. Regime Switching

Lemma B.3 shows that, for any $t \in [0, T)$, the map $x \mapsto -\log \tilde{p}_t(x)$ is weakly convex, having weak convexity profile satisfying $\kappa_{-\log \tilde{p}_t}(r) \geq C_t$, with $C_t$ as in (20). This implies that weak log-concavity of the data distribution evolves into log-concavity over time and that the drift of the time-reversed OU process alternates between contractive and non-contractive regimes.

**Log-concavity.** Note that the constant $C_t + 1$ represents the (weak) log-concavity constant of the density $\overrightarrow{p}_t$ associated with the process $\overrightarrow{X}_t$. The estimate for $(C_t + 1)$ is coherent with the intuition we have on the SGMs. Indeed, for $t = 0$, it matches the weak log-concavity constant of $\pi_{\text{data}}$, as $C_0 + 1 = \alpha - M$, and, for $t \to +\infty$, it matches the log-concavity constant of $\pi_\infty$, as

$$C_t + 1 = \frac{\alpha}{\alpha + (1-\alpha)e^{-2t}} - \frac{e^{-2t}}{(\alpha + (1-\alpha)e^{-2t})^2} M \longrightarrow 1, \quad \text{for} \quad t \to +\infty.$$

Computing when $C_t + 1 > 0$, we have that two regimes appear:

- $\overrightarrow{p}_t$ is only weakly log-concave for $t \in [0, \xi(\alpha, M)]$;

- $\overrightarrow{p}_t$ is log-concave for $t \in [\xi(\alpha, M), T]$,

with

$$\xi(\alpha, M) := \begin{cases} \log\left(\sqrt{\frac{\alpha^2 + M - \alpha}{\alpha^2}}\right) \wedge T, & \text{if } \alpha - M < 0, \\ 0, & \text{otherwise .} \end{cases} \tag{22}$$

When $\pi_{\text{data}}$ is only weakly log-concave, i.e., when $\alpha - M < 0$, we have that $\frac{\alpha^2 + M - \alpha}{\alpha^2} > 1$. Thus, $\xi(\alpha, M) > 0$ and two distinct regimes are present. Whereas, when the initial distribution $\pi_{\text{data}}$ is log-concave, i.e., when $\alpha - M \geq 0$, we have that $\xi(\alpha, M) = 0$ and $\overrightarrow{p}_t$ is log-concave in the whole interval $[0, T]$.

**Contractivity properties of the time-reversal process.** Denote by $T^\star$ the following time

$$T^\star := \inf\left\{t \in [0, T] \ : \ 2\kappa_{-\log \tilde{p}_t} + 1 \geq 0\right\}, \tag{23}$$

defining $\inf \varnothing := T$. From equation (5), we see that the time $T^\star$ corresponds to the first moment beyond which the drift $b_t$, used to define the time reversal process, defined in (6), is no longer contractive. Indeed, from the very definition of $\kappa_{-\log \tilde{p}_t}$, we get that

$$(b_t(x) - b_t(x))^\top (x - y) \leq -(2\kappa_{-\log \tilde{p}_t}(\|x - y\|) + 1)\|x - y\|^2 ,$$

for $t \in [0, T]$ and $x, y \in \mathbb{R}^d$. Also, denote by

$$T(\alpha, M, \rho) := \inf\left\{t \in [0, T] \ : \ 2C_t + 1 \geq \rho\right\},$$

for $\rho \in [0, 1)$, with $\inf \varnothing := T$. From (19), we have that $T^\star \leq T(\alpha, M, \rho)$, for any $\rho \in [0, 1)$. This means that two regimes are present in the time interval $[0, T]$:

- $b_t(x)$ is not (necessarily) contractive, for $t \in [0, T(\alpha, M, 0)]$.

- $b_t(x)$ is contractive, for $t \in [T(\alpha, M, 0), T]$;

We introduce $T(\alpha, M, \rho)$ as a quantitative explicit bound for $T^\star$. Indeed, a straightforward computation shows that

$$T(\alpha, M, \rho) = T - \eta(\alpha, M, \rho), \tag{24}$$

with

$$\eta(\alpha, M, \rho) := \begin{cases} \frac{1}{2} \log \left( \frac{M+\rho\alpha(1-\alpha)}{(1-\rho)\alpha^2} + \sqrt{\frac{(1+\rho)(1-\alpha)^2}{(1-\rho)\alpha^2} + \left( \frac{M+\rho\alpha(1-\alpha)}{(1-\rho)\alpha^2} \right)^2} \right) \wedge T, & \text{if } 2\alpha - 2M - 1 < 0, \\ 0, & \text{otherwise}. \end{cases} \tag{25}$$

Note that the condition

$$\frac{M + \rho\alpha(1 - \alpha)}{(1 - \rho)\alpha^2} + \sqrt{\frac{(1 + \rho)(1 - \alpha)^2}{(1 - \rho)\alpha^2} + \left( \frac{M + \rho\alpha(1 - \alpha)}{(1 - \rho)\alpha^2} \right)^2} > 1$$

is equivalent to the condition $2\alpha - 2M - 1 < 0$. This means that $T(\alpha, M, \rho)$ is well-defined and $T(\alpha, M, \rho) \in [0, T]$.

This regime shift is a key element in the effectiveness of the SGMs. SDEs with contractive flows exhibit advantageous properties related to efficiency guarantees (see, *e.g.*, Dalalyan, 2017; Durmus & Moulines, 2017; Cheng et al., 2018; Dwivedi et al., 2019; Shen & Lee, 2019; Cao et al., 2020; Mou et al., 2021; Li et al., 2021) that we can exploit in the large-time regime.

### B.4. Regularity in space of the modified score function

**Proposition B.4.** *Assume that Assumption H1 holds and fix $t \in [0, T)$. Then, it holds*

$$\sup_{x \in \mathbb{R}^d} \left\| \nabla^2 \log \tilde{p}_t(x) \right\| \le L_t \le L, \tag{26}$$

*where*

$$L_t = \max \left\{ \min \left\{ \frac{1}{1 - \mathrm{e}^{-2(T-t)}} \, ; \, \mathrm{e}^{2(T-t)} L_U \right\} \, ; \, -(C_t + 1) \right\} + 1, \quad L = \max\{1/\alpha; 1\}(2 + L_U). \tag{27}$$

*In particular, the modified score function $x \mapsto \nabla \log \tilde{p}_t(x)$ is $L_t$-Lipschitz, i.e.,*

$$\left\| \nabla \log \tilde{p}_t(x) - \nabla \log \tilde{p}_t(y) \right\| \le L_t \left\| x - y \right\|, \tag{28}$$

*for any $x, y \in \mathbb{R}^d$.*

*Proof. Step 1: Upper bound on $\nabla^2 \log \overrightarrow{p}_t$.* Recall that the transition density associated to the Orstein–Uhlenbeck semigroup is given by

$$q_t(x, y) = \frac{1}{(2\pi(1 - \mathrm{e}^{-2t}))^{d/2}} \exp \left( -\frac{\| y - \mathrm{e}^{-t}x \|^2}{2(1 - \mathrm{e}^{-2t})} \right). \tag{29}$$

Therefore, $\overrightarrow{p}_t$ is given by

$$\overrightarrow{p}_t(y) = \int \frac{1}{(2\pi(1 - \mathrm{e}^{-2t}))^{d/2}} \exp \left( -\frac{\| y - \mathrm{e}^{-t}x \|^2}{2(1 - \mathrm{e}^{-2t})} - U(x) \right) \mathrm{d}x.$$

This means that $\overrightarrow{p}_t$ is the density of the sum of two independent random variables $Y_t^1 + Y_t^0$ of density respectively $q_{0,t}$ and $q_{1,t}$, such that

$$q_{0,t}(x) := \mathrm{e}^{td}\mathrm{e}^{-U\left(\mathrm{e}^t x\right)} = \mathrm{e}^{-\phi_{0,t}(x)},$$

$$q_{1,t}(x) := \frac{1}{(2\pi(1 - \mathrm{e}^{-2t}))^{d/2}} \exp \left( -\frac{\| x \|^2}{2(1 - \mathrm{e}^{-2t})} \right) = \mathrm{e}^{-\phi_{1,t}(x)},$$

for two functions $\phi_{0,t}$ and $\phi_{1,t}$. From the proof of Saumard & Wellner (Proposition 7.1, 2014), we get

$$\nabla^2 \left( -\log \overrightarrow{p}_t \right)(x) = -\mathrm{Var}(\nabla\phi_{0,t}(X_0)|X_0 + X_1 = x) + \mathbb{E}[\nabla^2\phi_{0,t}(X_0)|X_0 + X_1 = x]$$
$$= -\mathrm{Var}(\nabla\phi_{1,t}(X_1)|X_0 + X_1 = x) + \mathbb{E}[\nabla^2\phi_{1,t}(X_1)|X_0 + X_1 = x].$$

From Bouchut et al. (2005, Lemma 2.2) and Assumption H1(i), we get that the one-sided Lipschitz assumption entails the following inequality over the Hessian of the log-density

$$\nabla^2 U \preccurlyeq L_U I_d.$$

Combining this with the fact that $\phi_{0,t}(x) = U(e^t x) + C$ (resp. $\phi_1 = 1/(2(1 - e^{-2t})) \|y\|^2 + C'$) for some positive constant $C$ (resp. $C'$), we obtain

$$\nabla^2\phi_{0,t} \preccurlyeq e^{2t} L_U \, I_d \quad \left( \text{resp. } \nabla^2\phi_{1,t} \preccurlyeq \frac{1}{1 - e^{-2t}} I_d \right),$$

which yields that

$$\nabla^2 \left( -\log \overrightarrow{p}_t \right)(x) \preccurlyeq \min\left\{ \frac{1}{1 - e^{-2t}}; e^{2t} L_U \right\} I_d. \tag{30}$$

*Step 2: Lower bound on* $\nabla^2 \log \overrightarrow{p}_t$. Using Lemma B.3, we get that (21) implies that

$$\nabla^2 \left( -\log \overrightarrow{p}_t \right)(x) \succcurlyeq -(C_t + 1)I_d. \tag{31}$$

Since the difference between $\nabla \log \overrightarrow{p}_t$ and $\nabla \log \tilde{p}_t$ is the linear function $x \mapsto -x$, we can take $L_t = \min\left\{ \frac{1}{1-e^{-2t}}; e^{2t} L_U \right\} + 1$. This implies that we can define $L_t$ as

$$L_t := \max\left\{ \min\left\{ \frac{1}{1 - e^{-2t}}; e^{2t} L_U \right\} + 1 \, ; \, -(C_t + 1) \right\} + 1.$$

*Step 3: Global bound in time.* If $C_0 + 1 = \alpha - M \geq 0$, we have that the initial distribution $\pi_{\mathrm{data}}$ is log-concave and, from Lemma B.3, $\overrightarrow{p}$ is log-concave in the whole interval $[0, T]$. In this case, the eigenvalues of the matrix $\nabla^2(-\log \overrightarrow{p}_t)$ are all positives and (30) provides a bound for $\|\nabla^2(-\log \overrightarrow{p}_t)\|_2$. Therefore, in this case a global bound in time for $L_t$ is

$$\sup_{t\in[0,\infty)} \left\{ \min\left\{ \frac{1}{1 - e^{-2t}}; e^{2t} L_U \right\} \right\} + 1 = 2 + L_U.$$

If $C_0 + 1 = \alpha - M < 0$, we also need to compute $\sup_{t\in[0,\infty)} -(C_t + 1)$. In this case, we have that

$$-(C_t + 1) = \frac{e^{-2t}}{(\alpha + (1-\alpha)e^{-2t})^2} M - \frac{\alpha}{\alpha + (1-\alpha)e^{-2t}}$$
$$= \frac{1}{\alpha(1 - e^{-2t}) + e^{-2t}} \left( \frac{e^{-2t} M}{\alpha + (1-\alpha)e^{-2t}} - \alpha \right)$$
$$= \frac{1}{\alpha(1 - e^{-2t}) + e^{-2t}} \left( \frac{M}{\alpha(e^{2t} - 1) + 1} - \alpha \right)$$
$$\leq \max\{1/\alpha; 1\} (M - \alpha).$$

Since $\alpha - M$ corresponds to the smallest eigenvalue of $\nabla^2 U$ and $L_U$ can be take equal to $\|\nabla^2 U\|_2$, in this case, we have that $M - \alpha \leq L_U$. Therefore, we can take as a global bound in time for $L_t$ the constant $L = \max\{1/\alpha; 1\}(2 + L_U)$.

$$\square$$

## C. Main Theorem

In this section we provide the formal version of Theorem 3.5 and we prove it.

**Theorem C.1.** *Suppose that Assumption H1 and H2 hold. Consider the discretization $\{t_k, 0 \le k \le N\}$ of $[0, T]$ of constant step size $h$ such that*

$$h < 2/9L^2 \,. \tag{32}$$

*Then, it holds that*

$$\mathcal{W}_2\left(\pi_{\text{data}}, \mathcal{L}(X_{t_N}^\star)\right) \le e^{3L\eta(\alpha, M, 9L^2 h/2)}\left[ e^{-T} \mathcal{W}_2\left(\pi_{\text{data}}, \pi_\infty\right) + 4\varepsilon\left(T(\alpha, M, 0)\right) \right.$$
$$\left. + \sqrt{2h}\left(B + 6L\sqrt{d}\right)\left(T(\alpha, M, 0)\right) \right],$$

*with $\eta(\alpha, M, 9L^2 h/2)$ as in (25), $L$ as in (27), and*

$$B := \sqrt{\mathrm{m}_2 + d} \,. \tag{33}$$

*Proof.* Recall the main regularity properties from Appendix B:

1. the modified score function $(t, x) \mapsto \nabla \log \tilde{p}_t(x)$ is $L_t$-Lipschitz in space and also $L$-Lipschitz in space uniformly in time, with $L_t$ and $L$ as in (26)-(27);

2. the function $(t, x) \mapsto -\log \tilde{p}_t(x)$ is $C_t$-weakly convex, with $C_t$ as in (20).

In the practical implementation of the algorithm, three successive approximations are made, generating three distinct sources of errors. To identify them, we introduce the following two processes.

- Let $(X_{t_k}^N)_{k=0}^N$ be the EM–approximation of the backward process (5) started at $X_0^N \sim \mathcal{L}(\overrightarrow{X}_T)$ and defined recursively on $[t_k, t_{k+1}]$ as

$$X_{t_{k+1}}^N = X_{t_k}^N + h_k\left(-X_{t_k}^N + 2\nabla \log \tilde{p}_{T-t_k}(X_{t_k}^N)\right) + \sqrt{2h_k}Z_k \,, \quad \text{for } t \in [t_k, t_{k+1}]\,,$$

with $\{Z_k\}_k$ a sequence of i.i.d. standard Gaussian random variables.

- Let $(X_{t_k}^\infty)_{k=0}^N$ be the EM–approximation of the backward process (5) started at $X_0^\infty \sim \pi_\infty$ and defined recursively on $[t_k, t_{k+1}]$ as

$$X_{t_{k+1}}^\infty = X_{t_k}^\infty + h_k\left(-X_{t_k}^\infty + 2\nabla \log \tilde{p}_{T-t_k}(X_{t_k}^\infty)\right) + \sqrt{2h_k}Z_k \,, \quad \text{for } t \in [t_k, t_{k+1}]\,.$$

We also recall that $(X_{t_k}^\star)_{k=0}^N$ defined in (8) denotes the process started at $X_0^\star \sim \pi_\infty$ and defined recursively on $[t_k, t_{k+1}]$ as

$$X_{t_{k+1}}^\star = X_{t_k}^\star + h_k\left(-X_{t_k}^\star + 2s_{\theta^\star}(T - t_k, X_{t_k}^\star)\right) + \sqrt{2h_k}Z_k \,, \quad \text{for } t \in [t_k, t_{k+1}]\,.$$

By an abuse of notation, we use $X^N$, $X^\infty$, and $X^\star$ to refer to both the discrete-time versions of these processes and their continuous-time interpolations. Applying the triangle inequality, we derive the following decomposition of the error bound

$$\mathcal{W}_2\left(\pi_{\text{data}}, \mathcal{L}(X_{t_N}^\star)\right)$$
$$\le \mathcal{W}_2\left(\mathcal{L}(\overleftarrow{X}_T), \mathcal{L}(X_{t_N}^N)\right) + \mathcal{W}_2\left(\mathcal{L}(X_{t_N}^N), \mathcal{L}(X_{t_N}^\infty)\right) + \mathcal{W}_2\left(\mathcal{L}(X_{t_N}^\infty), \mathcal{L}(X_{t_N}^\star)\right).$$

In the following, we establish separate bounds for each term, contributing to the overall convergence bound, based on the preceding analysis.

**Bound on** $\mathcal{W}_2\left(\mathcal{L}(\overleftarrow{X}_T), \mathcal{L}(X_{t_N}^N)\right)$**.** Consider the synchronous coupling between $(\overleftarrow{X}_t)_{t\in[0,T]}$ and the continuous-time interpolation of $(X_{t_k}^N)_{k=0}^N$ with the same initialization, $i.e.$ use the same Brownian motion to drive the two processes and set $\overleftarrow{X}_0 = X_0^N$. Then, it holds

$$\mathcal{W}_2\left(\mathcal{L}(\overleftarrow{X}_T), \mathcal{L}(X_{t_N}^N)\right) \leq \left\|\overleftarrow{X}_T - X_T^N\right\|_{L_2}.$$

To upper bound the r.h.s., we estimate $\|\overleftarrow{X}_{t_{k+1}} - X_{t_{k+1}}^N\|_{L_2}$ by means of $\|\overleftarrow{X}_{t_k} - X_{t_k}^N\|_{L_2}$, and develop the recursion.

Fix $0 < \epsilon < h$ and, with abuse of notation, use $T$ to denote $T - \epsilon$ and $N$ to denote $(T - \epsilon)/h$. This measure is necessary to enable the application of Conforti et al. (2023a, Proposition 2) and Proposition B.4 later on. As we considered the synchronous coupling between $\overleftarrow{X}$ and $X^N$, we get

$$\overleftarrow{X}_{t_{k+1}} - X_{t_{k+1}}^N$$
$$= \overleftarrow{X}_{t_k} - X_{t_k}^N + \int_{t_k}^{t_{k+1}} \left\{ -\left(\overleftarrow{X}_t - X_{t_k}^N\right) + 2\left(\nabla \log \tilde{p}_{T-t}\left(\overleftarrow{X}_t\right) - \nabla \log \tilde{p}_{T-t_k}\left(X_{t_k}^N\right)\right)\right\} \mathrm{d}t.$$

Using triangle inequality, we obtain

$$\left\|\overleftarrow{X}_{t_{k+1}} - X_{t_{k+1}}^N\right\|_{L_2}$$
$$\leq \left\|\overleftarrow{X}_{t_k} - X_{t_k}^N \right.$$
$$\left. + \int_{t_k}^{t_{k+1}} \left\{ -\left(\overleftarrow{X}_{t_k} - X_{t_k}^N\right) + 2\left(\nabla \log \tilde{p}_{T-t_k}\left(\overleftarrow{X}_{t_k}\right) - \nabla \log \tilde{p}_{T-t_k}\left(X_{t_k}^N\right)\right)\right\} \mathrm{d}t \right\|_{L_2} \tag{34}$$
$$+ \left\|\int_{t_k}^{t_{k+1}} \left\{ -\left(\overleftarrow{X}_t - \overleftarrow{X}_{t_k}\right) + 2\left(\nabla \log \tilde{p}_{T-t}\left(\overleftarrow{X}_t\right) - \nabla \log \tilde{p}_{T-t_k}\left(\overleftarrow{X}_{t_k}\right)\right)\right\} \mathrm{d}t \right\|_{L^2}$$
$$=: A_{1,k} + A_{2,k},$$

*Bound of $A_{1,k}$.* The first term of r.h.s. of (34) can be bounded as

$$A_{1,k}^2 = \left\|\overleftarrow{X}_{t_k} - X_{t_k}^N\right\|_{L_2}^2$$
$$+ h^2 \left\|-\left(\overleftarrow{X}_{t_k} - X_{t_k}^N\right) + 2\left(\nabla \log \tilde{p}_{T-t_k}\left(\overleftarrow{X}_{t_k}\right) - \nabla \log \tilde{p}_{T-t_k}\left(X_{t_k}^N\right)\right)\right\|_{L_2}^2$$
$$+ 2h\mathbb{E}\left[\left(\overleftarrow{X}_{t_k} - X_{t_k}^N\right)^\top \left(-\left(\overleftarrow{X}_{t_k} - X_{t_k}^N\right)\right.\right.$$
$$\left.\left.+ 2\left(\nabla \log \tilde{p}_{T-t_k}\left(\overleftarrow{X}_{t_k}\right) - \nabla \log \tilde{p}_{T-t_k}\left(X_{t_k}^N\right)\right)\right)\right].$$

Since $h$ satisfies (32), using Lemma D.2, we have that

$$h \leq \frac{2C_{T-t} + 1}{9L^2} \wedge 1, \qquad \text{for } t \in \left[0, T - \eta(\alpha, M, 9L^2h/2)\right],$$

with $\eta(\alpha, M, \rho)$ as in (25).

Define $N_h := \sup\left\{k \in \{0, ..., N\} : t_k \leq T - \eta(\alpha, M, 9L^2h/2)\right\}$. Using the regularity properties of the score function from Proposition B.4 and Lemma B.3, we have

$$A_{1,k} \leq \begin{cases} \left\|\overleftarrow{X}_{t_k} - X_{t_k}^N\right\|_{L_2}\left(1 + h^2\left(2L_{T-t_k} + 1\right)^2 - 2h\left(2C_{T-t_k} + 1\right)\right)^{1/2}, & \text{for } k < N_h, \\ \left\|\overleftarrow{X}_{t_k} - X_{t_k}^N\right\|_{L_2}\left(1 + h^2\left(2L_{T-t_k} + 1\right)^2 + 2h\left(2L_{T-t_k} + 1\right)\right)^{1/2}, & \text{for } k \geq N_h \end{cases} \tag{35}$$
$$:= \delta_k \left\|\overleftarrow{X}_{t_k} - X_{t_k}^N\right\|_{L_2}. \tag{36}$$

*Bound of $A_{2,k}$.* Given the definition (6) of the backward drift $b_t$, Jensen's inequality implies

$$A_{2,k}^2 = \left\| \int_{t_k}^{t_{k+1}} \left\{ b_{T-t}(\overleftarrow{X}_t) - b_{T-t_k}(\overleftarrow{X}_{t_k}) \right\} \mathrm{d}t \right\|_{L^2}^2 = \mathbb{E}\left[ \left\| \int_{t_k}^{t_{k+1}} \left\{ b_{T-t}(\overleftarrow{X}_t) - b_{T-t_k}(\overleftarrow{X}_{t_k}) \right\} \mathrm{d}t \right\|^2 \right]$$

$$\leq h \int_{t_k}^{t_{k+1}} \mathbb{E}\left[ \left\| b_{T-t}(\overleftarrow{X}_t) - b_{T-t_k}(\overleftarrow{X}_{t_k}) \right\|^2 \right] \mathrm{d}t .$$

Applying Itô's formula and Conforti et al. (Proposition 2, 2023a), we obtain

$$\mathrm{d}b_{T-t}\left(\overleftarrow{X}_t\right) = -\mathrm{d}\overleftarrow{X}_t + 2\,\mathrm{d}\left(\nabla \log \tilde{p}_{T-t}\left(\overleftarrow{X}_t\right)\right)$$

$$= \left\{ \overleftarrow{X}_t - 2\nabla \log \tilde{p}_{T-t}\left(\overleftarrow{X}_t\right) + 2\nabla \log \tilde{p}_{T-t}\left(\overleftarrow{X}_t\right) \right\} \mathrm{d}t + \sqrt{2}\left(1 + 2\nabla^2 \log \tilde{p}_{T-t}\left(\overleftarrow{X}_t\right)\right) \mathrm{d}B_t$$

$$= \overleftarrow{X}_t \mathrm{d}t + \sqrt{2}\left(1 + 2\nabla^2 \log \tilde{p}_{T-t}\left(\overleftarrow{X}_t\right)\right) \mathrm{d}B_t .$$

Therefore, using Jensen's inequality, Itô's isometry, Proposition B.4 and Lemma D.1, we get

$$\mathbb{E}\left[ \left\| b_{T-t}\left(\overleftarrow{X}_t\right) - b_{T-t_k}(\overleftarrow{X}_{t_k}) \right\|^2 \right] = \mathbb{E}\left[ \left\| \int_{t_k}^t \overleftarrow{X}_s \mathrm{d}s + \sqrt{2} \int_{t_k}^t \left(1 + 2\nabla^2 \log \tilde{p}_{T-s}\left(\overleftarrow{X}_s\right)\right) \mathrm{d}B_s \right\|^2 \right]$$

$$\leq \mathbb{E}\left[ 2\int_{t_k}^t \left\| \overleftarrow{X}_s \right\|^2 \mathrm{d}s + 4\int_{t_k}^t \left\| \mathbb{I} + 2\nabla^2 \log \tilde{p}_{T-s}\left(\overleftarrow{X}_s\right) \right\|_{\mathrm{Fr}}^2 \mathrm{d}s \right]$$

$$\leq \mathbb{E}\left[ 2\int_{t_k}^t \left\| \overleftarrow{X}_s \right\|^2 \mathrm{d}s + 8\int_{t_k}^t d\left\{ 1 + 4\left\| \nabla^2 \log \tilde{p}_{T-s}\left(\overleftarrow{X}_s\right) \right\|_{\mathrm{op}}^2 \right\} \mathrm{d}s \right]$$

$$\leq 2\int_{t_k}^{t_{k+1}} \mathbb{E}\left[ \left\| \overleftarrow{X}_s \right\|^2 \right] \mathrm{d}s + 8hd\left(1 + 4L^2\right)$$

$$\leq 2hB^2 + 8hd\left(1 + 4L^2\right) .$$

where we have used that

$$\|A\|_{\mathrm{Fr}}^2 \leq d\|A\|_{\mathrm{op}}^2 , \qquad \text{for a symmetric matrix } A \in \mathbb{R}^{d \times d} ,$$

with $\| \cdot \|_{\mathrm{Fr}}$ (resp. $\| \cdot \|_{\mathrm{op}}$) the Frobenius norm (operatorial norm) of a matrix.

Consequently, we have that

$$A_{2,k} \leq \left( h \int_{t_k}^{t_{k+1}} \left\{ 2hB^2 + 8hd\left(1 + 4L^2\right) \right\} \mathrm{d}t \right)^{1/2} \leq \sqrt{2h}\left(B + 2(2L+1)\sqrt{d}\right)h \leq \sqrt{2h}\left(B + 6L\sqrt{d}\right)h .$$

Finally, we obtain

$$\left\| \overleftarrow{X}_{t_{k+1}} - X_{t_{k+1}}^N \right\|_{L_2} \leq \delta_k \left\| \overleftarrow{X}_{t_k} - X_{t_k}^N \right\|_{L_2} + \sqrt{2h}\left(B + 6L\sqrt{d}\right)h . \tag{37}$$

Developing the recursion (37) and using the fact that $\overleftarrow{X}_0 = X_0^N$, we get

$$\left\| \overleftarrow{X}_T - X_T^N \right\|_{L_2} \leq \left\| \overleftarrow{X}_0 - X_0^N \right\|_{L_2} \prod_{\ell=0}^{N-1} \delta_\ell + \sqrt{2h}\left(B + 6L\sqrt{d}\right)h \sum_{k=0}^{N-1} \prod_{\ell=k}^{N-1} \delta_\ell$$

$$= \sqrt{2h}\left(B + 6L\sqrt{d}\right)h \sum_{k=0}^{N-1} \prod_{\ell=k}^{N-1} \delta_\ell .$$

Lemma D.2 yields that $\delta_k \leq 1$ for $k < N_h$. Whereas, for $k \geq N_h$, Proposition B.4 yields

$$\delta_k = \left(1 + h^2\left(2L_{T-t_k} + 1\right)^2 + 2h\left(2L_{T-t_k} + 1\right)\right)^{1/2} \leq 1 + h(2L+1) \leq 1 + 3Lh .$$

Combining the previous remarks, we have

$$
\begin{aligned}
\sum_{k=0}^{N-1}\prod_{\ell=k}^{N-1}\delta_\ell &= \sum_{k=0}^{N_h-1}\prod_{\ell=k}^{N_h-1}\delta_\ell \times \prod_{\ell=N_h}^{N-1}\delta_\ell + \sum_{k=N_h}^{N-1}\prod_{\ell=k}^{N-1}\delta_\ell \\
&\leq N_h\prod_{\ell=N_h}^{N-1}\delta_\ell + \sum_{k=N_h}^{N-1}\prod_{\ell=k}^{N-1}\delta_\ell \\
&\leq N_h\left(1+3Lh\right)^{N-N_h} + \sum_{k=N_h}^{N-1}\left(1+3Lh\right)^{N-k-1} \\
&\leq N_h\left(1+3Lh\right)^{N-N_h} + \sum_{k=0}^{N-N_h-1}\left(1+3Lh\right)^{k} \\
&= N_h\left(1+3Lh\right)^{N-N_h} + \frac{1-\left(1+3Lh\right)^{N-N_h}}{1-\left(1+3Lh\right)} \\
&\leq \left(N_h + \frac{1}{3Lh}\right) \times \left(1+3Lh\right)^{N-N_h} \\
&\leq \left(\frac{T-\eta(\alpha,M,9L^2h/2)}{h} + \frac{1}{3Lh}\right) e^{3L\eta(\alpha,M,9L^2h/2)}.
\end{aligned}
$$

Putting all these inequalities together, we get

$$
\begin{aligned}
\left\|\overleftarrow{X}_T - X_T^N\right\|_{L_2}^2 &\leq \sqrt{2h}\left(B+6L\sqrt{d}\right)\left(T-\eta(\alpha,M,9L^2h/2)+\frac{1}{3L}\right)e^{3L\eta(\alpha,M,9L^2h/2)} \\
&\leq \sqrt{2h}\left(B+6L\sqrt{d}\right)\left(T(\alpha,M,0)+\frac{1}{3L}\right)e^{3L\eta(\alpha,M,9L^2h/2)},
\end{aligned}
$$

where, in the last inequality, we have used the fact that

$$
T(\alpha,M,9L^2h/2) = T-\eta(\alpha,M,9L^2h/2) \leq T-\eta(\alpha,M,0) = T(\alpha,M,0).
$$

Recalling that we have used $T$ to denote $T-\epsilon$, we get

$$
\begin{aligned}
\left\|\overleftarrow{X}_{T-\epsilon} - X_{T-\epsilon}^N\right\|_{L_2}^2 &\leq \sqrt{2h}\left(B+6L\sqrt{d}\right)\left(T(\alpha,M,0)+\frac{1}{3L}\right)e^{3L\eta(\alpha,M,9L^2h/2)} \\
&\leq \sqrt{2h}\left(B+6L\sqrt{d}\right)\left(T(\alpha,M,0)+\frac{1}{3L}\right)e^{3L\eta(\alpha,M,9L^2h/2)}.
\end{aligned}
$$

Letting $\epsilon$ going to zero and using Fatou's lemma, we obtain

$$
\left\|\overleftarrow{X}_T - X_T^N\right\|_{L_2}^2 \leq \sqrt{2h}\left(B+6L\sqrt{d}\right)\left(T(\alpha,M,0)+\frac{1}{3L}\right)e^{3L\eta(\alpha,M,9L^2h/2)}.
$$

**Bound on $\mathcal{W}_2\left(\mathcal{L}(X_{t_N}^N),\mathcal{L}(X_{t_N}^\infty)\right)$.** Consider the synchronous coupling between the continuous-time interpolations of $(X_{t_k}^N)_{k=0}^N$ and $(X_{t_k}^\infty)_{k=0}^N$ with initialization satisfying

$$
\mathcal{W}_2(\pi_\infty,\mathcal{L}(\overrightarrow{X}_T)) = \|X_0^\infty - X_0^N\|_{L^2}.
$$

Then, we have that

$$
\mathcal{W}_2\left(\mathcal{L}(X_{t_N}^N),\mathcal{L}(X_{t_N}^\infty)\right) \leq \left\|X_T^N - X_T^\infty\right\|_{L_2}.
$$

Fix $0 < \epsilon < h$ and, with abuse of notation, use $T$ to denote $T - \epsilon$ and $N$ to denote $(T - \epsilon)/h$. This measure is necessary to enable the application of Proposition B.4 later on. As previously done, we aim to develop a recursion over the time intervals $[t_{k+1}, t_k]$. To this end, note that, based on the definitions of $(X_t^N)_{t \in [0,T]}$ and $(X_t^\infty)_{t \in [0,T]}$, and using the triangle inequality, we have that

$$
\left\| X_{t_{k+1}}^N - X_{t_{k+1}}^\infty \right\|_{L_2}
$$
$$
= \left\| X_{t_k}^N - X_{t_k}^\infty + \int_{t_k}^{t_{k+1}} \left\{ -\left( X_{t_k}^N - X_{t_k}^\infty \right) + 2\left( \nabla \log \tilde{p}_{T-t_k}\left( X_{t_k}^N \right) - \nabla \log \tilde{p}_{T-t_k}\left( X_{t_k}^\infty \right) \right) \right\} dt \right\|_{L_2} . \tag{38}
$$

Proceeding as in the previous bound, we get

$$
\left\| X_T^N - X_T^\infty \right\|_{L_2} \leq \left\| X_0^N - X_0^\infty \right\|_{L_2} \prod_{\ell=0}^{N-1} \delta_\ell ,
$$

with $\delta_k$ defined as in (36) We bound the first factor in the following (by now) standard (see, *e.g.*, the proof of Proposition C.2, Strasman et al., 2024)

$$
\left\| X_0^N - X_0^\infty \right\|_{L_2} = \mathcal{W}_2(\pi_\infty, \mathcal{L}(\overrightarrow{X}_T)) \leq e^{-T} \mathcal{W}_2(\pi_{\text{data}}, \pi_\infty) .
$$

To bound the second factor, we proceed as in the bound of $\mathcal{W}_2\left( \mathcal{L}(\overleftarrow{X}_T), \mathcal{L}(X_{t_N}^N) \right)$ and get

$$
\prod_{\ell=0}^{N-1} \delta_\ell \leq \prod_{\ell=N_h}^{N-1} \delta_\ell \leq (1 + h3L)^{N-N_h} \leq e^{3L\eta(\alpha, M, 9L^2 h/2)} .
$$

Putting these inequalities together, we get

$$
\left\| X_T^N - X_T^\infty \right\|_{L_2} \leq e^{-T} e^{3L\eta(\alpha, M, 9L^2 h/2)} \mathcal{W}_2(\pi_{\text{data}}, \pi_\infty) .
$$

Recalling that we have used $T$ to denote $T - \epsilon$, we let $\epsilon$ going to zero and conclude using Fatou's lemma as in the bound of $\mathcal{W}_2\left( \mathcal{L}(\overleftarrow{X}_T), \mathcal{L}(X_{t_N}^N) \right)$.

**Bound on $\mathcal{W}_2\left( \mathcal{L}(X_{t_N}^\infty), \mathcal{L}(X_{t_N}^\star) \right)$.** Consider the synchronous coupling between the continuous-time interpolation of $(X_{t_k}^\infty)_{k=0}^N$ and $(X_{t_k}^\star)_{k=0}^N$, with the same initialization, *i.e.* use the same Brownian motion to drive the two processes and set $X_0^\infty = X_0^\star$. Then, it holds

$$
\mathcal{W}_2\left( \mathcal{L}(X_{t_N}^\infty), \mathcal{L}(X_{t_N}^\star) \right) \leq \left\| X_T^\infty - X_T^\star \right\|_{L_2} .
$$

Fix $0 < \epsilon < h$ and, with abuse of notation, use $T$ to denote $T - \epsilon$ and $N$ to denote $(T - \epsilon)/h$. This measure is necessary to enable the application of Proposition B.4 later on. As done in the previous bounds, we aim to develop a recursion over the time intervals $[t_{k+1}, t_k]$. To this end, using the triangle inequality, we get

$$
\left\| X_{t_{k+1}}^\infty - X_{t_{k+1}}^\star \right\|_{L_2}
$$
$$
= \left\| X_{t_k}^\infty - X_{t_k}^\star + \int_{t_k}^{t_{k+1}} \left\{ -\left( X_{t_k}^\infty - X_{t_k}^\star \right) + 2\left( \nabla \log \tilde{p}_{T-t_k}\left( X_{t_k}^\infty \right) - \tilde{s}_{\theta^\star}\left( T - t_k, X_{t_k}^\star \right) \right) \right\} dt \right\|_{L_2}^2
$$
$$
\leq \left\| X_{t_k}^\infty - X_{t_k}^\star + \int_{t_k}^{t_{k+1}} \left\{ -\left( X_{t_k}^\infty - X_{t_k}^\star \right) + 2\left( \nabla \log \tilde{p}_{T-t_k}\left( X_{t_k}^\infty \right) - \nabla \log \tilde{p}_{T-t_k}\left( X_{t_k}^\star \right) \right) \right\} dt \right\|_{L_2} \tag{39}
$$
$$
+ 4\left\| \int_{t_k}^{t_{k+1}} \left( \nabla \log \tilde{p}_{T-t_k}\left( X_{t_k}^\star \right) - \tilde{s}_{\theta^\star}\left( T - t_k, X_{t_k}^\star \right) \right) dt \right\|_{L_2}
$$
$$
= B_{1,k} + B_{2,k} .
$$

Following the same reasoning of the previous sections, we have that

$$B_{1,k} \leq \delta_k \left\| X_{t_k}^\infty - X_{t_k}^\star \right\|_{L_2},$$

with $\delta_k$ defined as in (36). Moreover, using Assumption H2, we have that $B_{2,k} \leq 4h\epsilon$.

Developing the recursion as in the previous sections and recalling that $X_0^\infty = X_0^\star$, we get

$$\|X_T^\infty - X_T^\star\|_{L_2}^2 \leq 4h\varepsilon \sum_{k=0}^{N-1} \prod_{\ell=k}^{N-1} \delta_\ell \leq 4\varepsilon \left( T(\alpha, M, 0) + \frac{1}{3L} \right) e^{3L\eta(\alpha, M, 9L^2 h/2)}.$$

Recalling that we have used $T$ to denote $T - \epsilon$, we let $\epsilon$ going to zero and conclude using Fatou's lemma as in the bound of $\mathcal{W}_2 \left( \mathcal{L}(\overleftarrow{X}_T), \mathcal{L}(X_{t_N}^N) \right)$.

$\square$

## D. Technical lemmata

**Lemma D.1.** *Assume that H1(iii) holds. Then, for $t \geq 0$,*

$$\sup_{0 \leq t \leq T} \left\| \overleftarrow{X}_t \right\|_{L_2} \leq \sup_{0 \leq t \leq T} \left( e^{-2(T-t)} \int_{\mathbb{R}^d} |x|^2 \pi_{\text{data}}(\mathrm{d}x) + \left( 1 - e^{-2(T-t)} \right) d \right)^{1/2} \leq B < \infty,$$

*for $B$ defined as in (33).*

*Proof.* Recall the following equality in law

$$\overrightarrow{X}_t = e^{-t} X_0 + \sqrt{(1 - e^{-2t})} G. \tag{40}$$

with $X_0 \sim \pi_{\text{data}}$, $G \sim \mathcal{N}(0, \mathbb{I})$, and $X_0$ and $G$ taken to be independent. Therefore, since $X_0$ and $G$ are independent and $\overleftarrow{X}_t$ has the same law of $\overrightarrow{X}_{T-t}$, we obtain

$$\mathbb{E}\left[ \left\| \overleftarrow{X}_t \right\|^2 \right] = \mathbb{E}\left[ \left\| \overrightarrow{X}_{T-t} \right\|^2 \right] \leq e^{-2(T-t)} \mathbb{E}\left[ \|X_0\|^2 \right] + \left( 1 - e^{-2(T-t)} \right) \mathbb{E}\left[ \|G\|^2 \right]$$

$$= e^{-2(T-t)} \int_{\mathbb{R}^d} |x|^2 \pi_{\text{data}}(\mathrm{d}x) + \left( 1 - e^{-2(T-t)} \right) d,$$

which concludes the proof.

$\square$

**Lemma D.2.** *Suppose that Assumption H1 holds. Consider the regular discretization $\{t_k, 0 \leq k \leq N\}$ of $[0, T]$ of constant step size $h$. Assume that $h > 0$ satisfies (32). Then, we have that*

$$h \leq \frac{2(2C_{T-t} + 1)}{9L^2} \wedge 1, \qquad \text{for } t \in \left[ 0, \eta(\alpha, M, 9L^2 h/2) \right], \tag{41}$$

*with $\eta(\alpha, M, \rho)$ as in (25). Moreover, for all $0 \leq k \leq N - 1$ such that $t_k \leq \eta(\alpha, M, 9L^2 h/2)$,*

$$0 < 1 + h^2 \left( 2L_{T-t_k} + 1 \right)^2 - 2h \left( 2C_{T-t_k} + 1 \right) \leq 1.$$

*Proof.* Firstly, note that if $h$ satisfies (32), then $\rho_h := 9L^2 h/2$ is such that $\rho_h \in [0, 1)$. Recalling the definitions (24), (25) of $T(\alpha, M, \rho)$ and $\eta(\alpha, M, \rho)$, we have that $2C_t + 1 \geq \rho_h$, for $t \geq T(\alpha, M, \rho_h)$. Otherwise said, we have

$$h \leq \frac{2(2C_t + 1)}{9L^2} \wedge 1, \qquad \text{for } t \in \left[ T(\alpha, M, 9L^2 h/2), T \right],$$

which is equivalent to

$$h \leq \frac{2(2C_{T-t} + 1)}{9L^2} \wedge 1, \qquad \text{for } t \in \left[0, \eta(\alpha, M, 9L^2h/2)\right].$$

Let $\epsilon_1$ be

$$\epsilon_1 := 1 + h^2 \left(2L_{T-t_k} + 1\right)^2 - 2h\left(2C_{T-t_k} + 1\right). \tag{42}$$

Completing the square, we obtain

$$\begin{aligned}
\epsilon_1 &= \left(1 - h\left(2L_{T-t_k} + 1\right)\right)^2 + 2h\left(2L_{T-t_k} + 1\right) - 2h\left(2C_{T-t_k} + 1\right) \\
&= \left(1 - h\left(2L_{T-t_k} + 1\right)\right)^2 + 4h\left(L_{T-t_k} - C_{T-t_k}\right).
\end{aligned}$$

The first term if the r.h.s. of the previous equality is a square, therefore always positive. The second term is always strictly positive, as $L_t \geq C_t$ for any $t$.

Secondly, we see that

$$\begin{aligned}
\epsilon_1 &= 1 + h^2\left(2L_{T-t_k} + 1\right)^2 - 2h\left(2C_{T-t_k} + 1\right) \\
&\leq 1 + h\left(h\left(2L_{T-t_k} + 1\right)^2 - 2\left(2C_{T-t_k} + 1\right)\right) \\
&\leq 1 + h\left(h\left(2L + 1\right)^2 - 2\left(2C_{T-t_k} + 1\right)\right).
\end{aligned}$$

From (41), we have that $\epsilon_1 \leq 1$, for $t_k \leq T - \eta(\alpha, M, 9L^2h/2)$. This concludes the proof. $\qquad \square$

