# OpenReview forum: "Beyond Log-Concavity and Score Regularity: Improved Convergence Bounds for Score-Based Generative Models in W2-distance"
_ICML.cc/2025/Conference — ICML 2025 poster_

### Official Review · Reviewer_Vb4a · 2025-03-10

**Overall Recommendation:** 4

**Summary:**

This paper takes a renewed look at analysing convergence of score-based generative models, in the Wasserstein-2 metric.   The explore guarantees under a weaker assumption than log-concavity, namely weak log-concavity (introduced in Conforti, 2023).    This enables the authors to obtain natural non-asymptotic convergence results, under more realistic assumptions.   Leveraging this machinery,   they recognize that the log-density of the forward process is the solution of an HJB equation, which is exploited to track the weak-log concavity over time - enabling them to characterise the transition between the log-concave regime (close to Gaussian) and the weak-log concave setting (close to data distribution).

# update after rebuttal
Based on the response, I maintain my score.

**Claims And Evidence:**

The paper is entirely performing detailed mathematical analysis of the convergence of score based models.   All the results and claims in the paper are either proved, using convincing arguments,  or referred to previous works.  There are no problematic claims.

**Essential References Not Discussed:**

I cannot think of any essential papers which are published and not cited here, or discussed.   What I had already mentioned is that there is some papers which generalise Prop 4.1 to gaussian mixtures with general covariance -  but I believe these are still preprints.

**Experimental Designs Or Analyses:**

There are no numerical experiments in this paper.

**Methods And Evaluation Criteria:**

The proposed methods are sensible.   As there are no numerical experiments, the evaluation criteria is not relevant here.

**Other Comments Or Suggestions:**

I've made most comments elsewhere.

**Other Strengths And Weaknesses:**

I think this is a clear contribution as it generally relaxes some key assumptions for obtaining non-asymptotic results in W2 for SBGMs.     I would have liked the author to explore a bit more what data-distributions actually satisfy weak log-concavity beyond the Gaussian / Gaussian Mixture case.

**Questions For Authors:**

No further questions.

**Relation To Broader Scientific Literature:**

There have been several prior works which derive similar non-asymptotic estimates for SBGMs using different metrics, ranging from classical Renyi / alpha divergences and TV,   Kullback-Liebler, and then Wasserstein - like in this paper.

In the former group, there is most notably,  [Block & Mroueh 2020], Valentin De Bortoli's paper "Convergence of denoising diffusion models under the manifold hypothesis",  and Chen and Chewi's "Sampling is as easy as learning the score...." paper.

In terms of Wasserstein-2, then de Bortoli, Heng, Doucet, et al's paper "Diffusion schrödinger bridge with applications to score-based generative modeling" and Lee, Lu, Tan's paper are the main contributions.

More recently, there was Sabanis et al's paper: "On diffusion-based generative models and their error bounds: The log-concave case with full convergence estimates",  Tang & Zhao's paper and Strassman et al, 2024.   The common thread in these works is the assumption of log-concavity of the Data Distribution, which is quite constraining and unrealistic.

**Theoretical Claims:**

I have checked the proof of Prop 4.1,  and all the theorems leading up to the main result, and the main result.    I have not followed references to results in other papers, and where I am not familiar, I have not dug too deeply.

The proofs look sound, the arguments look sensible.   I would add a few comments:

1. Prop 4.1 -- there have been recent works which establish log lipschitness of the score for a general mixed gaussian, even with non-isotropic covariances.    Can this result be generalised to that setting?   It would be nice to see that, or an acknowledgement that it is possible, or not.
2. An early, key assumption is the reformulation of equation (4) into (5) -- this is not a new assumption,  tracing back into one of Durmus' papers, and one of Cattiaux's earlier than that...   However, it is a non-trivial assumption that the initial distribution is ac with respect to pi^infinity.    The value of this assumption is clear, but the implications in terms of how limiting it is, is not.

3. In terms of style - there is no page limit on the supplemetnary information - so why not provide a bit more information, e.g. the HJB is crucial to the "once weak-log-concave, always weak log-concave" argument which is used in the main result.    Can a little bit more details be provided in this text, to better contextualise this paper?

4. The sqrt(dh)T contribution which arises from the EM discretisation is noteworthy,  but it really doesn't appear obvious from the main proof?   Is there a typo leading to equation (35), e.g. should the integrand be inside the square root?

---

> ### Author Rebuttal · Authors · 2025-03-30
>
> We appreciate the reviewer's insightful comments and supportive review.
>
> **On Gaussian Mixtures.** As highlighted in Remark 4.2, the generalization of Proposition 4.1 to Gaussian mixtures with non-isotropic components is indeed straightforward. It involves considering bounds based on the minimum or maximum eigenvalues of the covariance matrices. This adjustment allows one to control the local Lipschitz constants of the score function accordingly.
>
> Nevertheless, we would be very grateful if the reviewer could point us to the specific references they had in mind regarding recent results on the log-Lipschitz regularity of the score for general Gaussian mixtures. We would be happy to acknowledge them and expand our discussion in the revised version, possibly providing further insights into the verification of our assumptions in these more general settings.
>
> **AC assumption.** Since $\pi_{\infty}$ is a Gaussian distribution, absolute continuity (AC) of $\pi_{\mathrm{data}}$ w.r.t. $\pi_{\infty}$ is equivalent to AC w.r.t. the Lebesgue measure. This condition is directly implied by our Assumption H1. This requirement is still quite mild. For instance, if $\pi_{\mathrm{data}}$ were supported on a lower-dimensional manifold, the smoothing effect of the forward process ensures that $p_t$ becomes absolutely continuous w.r.t.\ $\pi_{\infty}$ for any $t > 0$, due to convolution with Gaussian noise. This makes the assumption broadly applicable and not restrictive in practice. A comment clarifying this point has been added in the revised version of the manuscript.
>
> **Link with the HJB equation.** We appreciate this insightful remark and have added a short subsection in the revised manuscript to elaborate on the connection between HJB equations, control theory, and SGMs. This addition draws on recent works (e.g., Berner et al., 2022; Zhang and Katsoulakis, 2023; Zhang et al., 2024a; Conforti et al., 2025), and aims to better contextualize our analysis by highlighting this rich and increasingly active intersection.
>
> **Typo.** We are grateful to the reviewer, as this helped us to spot a typo that has now been corrected. The $\sqrt{hd}T$-dependence remains valid.
>
> **On weakly log-concave distributions.** A notable class of weakly log-concave distributions arises from the convolution of distributions supported on lower-dimensional manifolds with a Gaussian kernel. This construction effectively smooths out the singularities, yielding a weakly log-concave distribution. This observation is also emphasized in Saremi et al. (2023). This insight may also help explain why early stopping techniques often perform better than directly modeling the data distribution: the intermediate distributions encountered during training are closer to weakly log-concave regimes, for which theoretical guarantees and sampling stability are more readily attainable. We have added a remark on this point in the revised version of the manuscript to clarify the broader applicability of our assumptions.

---

> > ### Comment · Reviewer_Vb4a · 2025-04-04
> >
> > I thank the reviewers for their careful responses. I will keep my score as is.

---

### Official Review · Reviewer_3brP · 2025-03-13

**Overall Recommendation:** 4

**Summary:**

The paper studies the properties of Score-based Generative Models (SGMs) beyond the conventional setting where the data  distribution $\pi_{\mathrm{data}}$ is log-concave and satisfies certain regularity conditions. The analysis is based on the three approximations that must be made to implement SGMs. First, one must initialize the backward process at a stationary distribution which is easy to sample from. Second, one must estimate the score function by training a model to minimize the score-matching loss. Third, the process must be discretized as the continuous SDE cannot be solved exactly.

The work focuses on Ornstein-Uhlenbeck Generative Models wherein the initial stationary distribution is chosen as a standard Gaussian. The analysis is carried out under the assumption that the data distribution has a density $\exp(-U(x))$ where $\nabla U$ satisfies a one-sided Lipschitz property and is weakly convex (a class which includes Gaussian mixtures) as well as a second, more technical condtion (cf. H2) which has appeared previously in the literature. The main theorem provides an estimate of how close the distribution outputted by the algorithm is to the true data distribution in Wasserstein distance is as a function of the size of the time interval [0,T], the coarseness of the discretization of this interval $h$, the dimension, a factor figuring assumption H2, and the distance of the standard Gaussian from the true distribution and is line with other literature and improves on the dependence in dimension.

The remainder of the article pertains to explaining the main ideas underlying the proof and discussing the case where the true data is generated according to a Gaussian mixture.

## update after rebuttal

The authors have answered the main questions that I have posed. As such, I maintain my initial score.

**Claims And Evidence:**

The claims in this submission appear reasonable to me, though I am not an expert in this topic.

**Essential References Not Discussed:**

N/A

**Experimental Designs Or Analyses:**

No experiments are performed in this work.

**Methods And Evaluation Criteria:**

No experiments are performed in this work.

**Other Comments Or Suggestions:**

Line 179 right column: that are at distance -> that are at a distance.
Line 239 right column: For sake of simplicity -> For the sake of simplicity.
Line 369 right column: These remarks yields that -> These remarks yield that

**Other Strengths And Weaknesses:**

I believe the paper is well-written and presents a useful relaxation of some of the conditions for score-based generative modelling. It is shown that the condition H1 is compatible with a Gaussian mixture model, but the question of whether or not H2 holds in this setting is not addressed. As such, I believe the condition H2 merits further discussion. Other than this, the outline in the main text helps to elucidate the proof technique and highlights how each level of approximation is handled.

**Questions For Authors:**

1. As noted previously, I wonder how easy condition H2 is to check in practice. It is a bit confusing, for instance, that Section 4 concentrates on the establishing H1 for the Gaussian Mixture, but says nothing about H2.

2. I wonder if it is feasible to empirically validate the derived theorem via some simulation. As the main contribution of this submission is theoretical, it would help to at least illustrate the findings at a numerical level to better motivate its applicability to real world problems.

**Relation To Broader Scientific Literature:**

I am not very familiar with the broader literature related to SGMs/sampling. It appears that most results are contingent on some form of log-concavity and regularity assumptions (e.g. log-sobolev) and hence this paper provides some relaxation of these conditions.

**Theoretical Claims:**

I did not verify the proofs in the supplement. Some arguments are provided in the main text and appear reasonable.

---

> ### Author Rebuttal · Authors · 2025-03-30
>
> We thank the reviewer for the thoughtful and positive feedback.
>
> **Assumption H2.** We emphasize that Assumption H2 is fully comparable to the standard estimation error assumptions widely used in the literature (see, e.g., Conforti et al., 2025; Benton et al., 2024; Chen et al., 2022a). In particular, our main result remains valid even if H2 is replaced by a more classical $L^2$-type control on the score approximation error of the forward process, provided that $\tilde s_{\theta^\star}(T - t_k, \cdot)$ is Lipschitz, uniformly in time. Importantly, Proposition B.2 ensures that this additional condition is not restrictive. Assumption H2 is therefore entirely consistent with standard theoretical frameworks, and its practical relevance is supported by well-established results. In particular, we refer to Appendix A of Chen et al. (2023), where this type of assumption is shown to hold in simple yet illustrative settings. We agree that a more detailed discussion of H2 would have added clarity. For this reason, we included a comment in the revised version of the manuscript to clarify this point.
>
> **Simulations.** We thank the reviewer for this valuable suggestion. We refer to Appendix E.3 of  Strasman et al. (2024) for an implementation of a similar bound. Our findings provide the theoretical foundation underlying these simulation results, and we believe a numerical study could be conducted by closely following the same methodological framework.
>
> Finally, we would like to thank the reviewer for the list of misprints and other small comments, which have been happily incorporated in the revised manuscript.

---

### Official Review · Reviewer_u274 · 2025-03-14

**Overall Recommendation:** 1

**Summary:**

This paper looks at showing that diffusion models can be quickly
sampled from with bounded W2 error.  There is a long line of work
showing this for TV or KL error, but converting those to W2 incurs a
significant penalty.  This paper shows a W2 bound directly, assuming
one-sided lipschitzness and weak log concavity.

## update after rebuttal

Given this response, I maintain my score.

(1) I asked for a direct comparison and you did not give one. At the very least you could give a bound for compact distributions, for comparison.

(2) But I'm not convinced you need any additional assumption like compactness. You have weak convexity of the score, which seems to imply subexponential tails, which seems like it should imply (TV => W2).

> a KL divergence bound does not, in general, imply a W2 bound without imposing additional, and often quite restrictive, assumptions, such as compact support,

Doesn't even a 2.1st moment work?

**Claims And Evidence:**

I'm very confused, because Theorem 3.4 is informal, and I *think*
Theorem D.1 is supposed to be the formal version of it.  But Theorem
D.1 is far, far weaker than Theorem 3.4.

Relative to Theorem 3.4, Theorem D.1 has:


- A leading e^{L_u eta} term, which in the mixture-of-Gaussian case is
  seems quite large if mu >> sigma

- ... at least I think, a lot of these terms seem dimensionally
  incorrect, like there should be some scaling that would be invariant
  (if I denominate x in feet rather than meters, the same process
  happens) but lots of terms would change.

- The sqrt(h) L_U d term is missing

- The sqrt(h) m_2 T term is missing

These set of errors really offend me, and make me recommend rejection
until the paper can be cleaned up.  When I first read Theorem 3.4, I
was really impressed and wanted to accept.  But partly I was really
impressed because it avoided any dependence on the things you would
get if you simply convert one of the KL/TV bounds (e.g. Benson et al)
into the W_2 setting; that would inherently give a dependence on the
moments.  But so does this method!  The paper just neglects to inform the
reader in the main body.

**Essential References Not Discussed:**

seemed fine

**Experimental Designs Or Analyses:**

none

**Methods And Evaluation Criteria:**

no new algorithm or evaluation criteria

**Other Comments Or Suggestions:**

none

**Other Strengths And Weaknesses:**

see above

**Questions For Authors:**

Is the result actually stronger than you would get by just converting
a KL bound like Benson?  Can you show this, for example, in the
balanced mixture of Gaussians case?


What can you do without assuming lipschitzness or weak log concavity?
Shouldn't you be able to say something like: the t-smoothed
distribution is R/t onesided lipschitz, to have fewer assumptions?

**Relation To Broader Scientific Literature:**

it's well positioned, trying to get a W2 bound directly.

**Theoretical Claims:**

see above

---

> ### Author Rebuttal · Authors · 2025-03-30
>
> We appreciate the detailed comments and constructive feedback. We highlight that, as stated in Theorem 3.4 and its accompanying discussion, this result indeed provides a bound up to a multiplicative constant depending on $\alpha, M, L_U$, and the second-order moment $m_2$ of $\pi_{\rm data}$, via the constant $B$ defined in Equation (31). Such a multiplicative constant includes all the terms listed by the reviewer. We thank them for the careful reading and address their concern in the revised manuscript as follows.
> - Theorem 3.4 now displays the **dependence on $m_2$**. This quantity no longer appears in the multiplicative constant; instead, its influence on the bound is now explicitly reflected. We have also emphasized that the multiplicative constant depends on the parameters $\alpha$, $M$, and $L_U$, with references to the explicit expression of the constant provided in the appendix.
> - We added a discussion on the leading term $e^{L_U \eta}$, highlighting that it reflects the intrinsic complexity of transforming a potentially complex, multimodal distribution into a unimodal Gaussian through an OU process. In particular, this term captures the challenge posed by large-scale structure or mode separation in the target distribution, as also highlighted in Saremi et al., 2023.
> - We particularly appreciated the observation regarding the $\sqrt{h} L_U d$ term. This helped us identify and correct a typo in the proof. The correct expression is actually $\sqrt{h d} L_U$, and this has now been fixed in the revised version of the manuscript.
>
> **Comparison with $W_2$-bounds.** We acknowledge that the approach in Benton et al. (2024) relies on different assumptions and methodologies, including exponentially decreasing step sizes and early stopping, while avoiding regularity assumptions on the data distribution. Given these differences, a direct side-by-side comparison is non-trivial.
>
> Our bound is directly comparable to those of Chen et al. (2023) and Conforti et al. (2025), as it exhibits the same dependencies on $h$, $d$, $\varepsilon$, $m_2$, and $T$. A key distinction, however, lies in the fact that our result is expressed in terms of the Wasserstein distance $W_2(\pi_{\text{data}}, \pi_\infty)$ rather than the KL divergence, which makes it more practical to estimate from samples—see, e.g., Strasman et al. (2024).
>
> **Regularity assumptions.** We very much agree that early stopping strategies could relax or remove the Lipschitz assumption. However, we believe that the explicit form of contraction property of the data distribution remains crucial to ensure the stability and convergence guarantees of our analysis.

---

> > ### Comment · Reviewer_u274 · 2025-04-05
> >
> > Thanks.  Then can you please give a direct comparison to Chen et al. or Conforti et al. ?  Like, a KL bound implies a TV bound implies a W2 bound over distributions of radius R; seems like that would avoid the exponential exp(L_U eta) dependence.
> >
> > I can see that mode separation can imply some loss, but I just don't buy that there should be an exponential dependence there.  Polynomial I would believe.

---

> > > ### Author Response · Authors · 2025-04-07
> > >
> > > We are grateful to the reviewer for their continued engagement and thoughtful input.
> > >
> > > We agree that drawing connections between KL, TV, and Wasserstein bounds is an interesting and valuable direction. However, a KL divergence bound does not, in general, imply a $\mathcal{W}_2$ bound without imposing additional, and often quite restrictive, assumptions, such as compact support, as suggested by the reviewer, Talagrand-type inequality or similar distribution tails' controls. We refer to Bolley \& Villani (2005) for further discussion on the conditions under which entropic bounds can be translated into trasport distances bounds.
> > >
> > > In our work, we deliberately avoided such strong conditions. In this sense, our goal was to fill a gap in the literature on SGMs under weak log-concavity conditions, providing a quantitative result where previously only heuristic or qualitative arguments were available (e.g., Saremi et al., 2023).
> > >
> > > We appreciate the reviewer’s remark regarding the exponential dependence, and we agree that improving this to a polynomial one under suitable conditions is both plausible and desirable. We view extending the theory under more refined assumptions as an exciting direction for future research.

---

### Official Review · Reviewer_gn4M · 2025-03-14

**Overall Recommendation:** 2

**Summary:**

This paper considers the sampling efficiency of diffusion models driven by the OU process in terms of Wasserstein distance under a new kind of "tilted" score-estimation assumption. The theorem only needs a weaker curvature condition for the potential function rather than the standard log-concavity. These conditions are shown to be satisfied by mixture of Gaussians. The paper also discussed the changing of curvature property over time for the forward density.

**Claims And Evidence:**

See the Section "Theoretical Claims".

**Essential References Not Discussed:**

To the best of my knowledge, technical references are well-discussed.

**Experimental Designs Or Analyses:**

N/A

**Methods And Evaluation Criteria:**

N/A

**Other Comments Or Suggestions:**

Minor suggestion: use different notations for the brownian motion in the forward and backward processes, e.g., in (3) and (4).

**Other Strengths And Weaknesses:**

### Minor issues
- Line 172: Maybe you miss the $\Sigma \Sigma^\top$ here after the $+$.

**Questions For Authors:**

N/A

**Relation To Broader Scientific Literature:**

The formulation of weak convexity and the PDE-based view in this papers come from
- Conforti, Giovanni, Alain Durmus, and Marta Gentiloni Silveri. "Score diffusion models without early stopping: finite fisher information is all you need." arXiv e-prints (2023): arXiv-2308.
- Conforti, Giovanni, Daniel Lacker, and Soumik Pal. "Projected Langevin dynamics and a gradient flow for entropic optimal transport." arXiv preprint arXiv:2309.08598 (2023).

These are the major direct technical predecessors of this paper.

There is also an extensive line of papers on the sampling efficiency of diffusion models, but the assumptions on score estimation in the current paper are not directly comparable with those in the literature, e.g.:
- Chen, Sitan, et al. "Sampling is as easy as learning the score: theory for diffusion models with minimal data assumptions." arXiv preprint arXiv:2209.11215 (2022).
- Li, Gen, et al. "Towards faster non-asymptotic convergence for diffusion-based generative models." arXiv preprint arXiv:2306.09251 (2023).

**Theoretical Claims:**

The proof of the main mixing theorem is largely correct, and it is great to have the Gaussian mixture example to illustrate the concept of weak convexity. But the major concern is:
- Since the score is tilted by standard Gaussian, this Assumption H2 is really not directly comparable with previous assumptions in the literature.
- If the authors do not provide some comparison between their Assumption H2, which is specialized to OU process and is tilted; and the standard $L^2$ estimation error assumption made in tons of previous (theoretical) papers on diffusion models, the significance of this paper is not clear.
- In particular, the claim that the dependence on $d$ "surpasses some earlier results" might not be very fair.

The risk decomposition seems to be standard and as expected. The authors should elaborate more on the view of $(t, x) \mapsto -\log \tilde{p}_{T-t}(x)$ as a solution to HJB, if it is indeed novel; instead of inserting Section 5 about the shifting of the curvature properties, which is relatively detached from the main mixing theorem of this paper.

Technically speaking, Section 5 is interesting, but the statement "not (necessarily) contractive" is really vague, since it is not clear whether it is the defect of the proof techniques or the inherent property of the backward process.

---

> ### Author Rebuttal · Authors · 2025-03-30
>
> We thank the reviewer for the time and effort taken to provide such detailed and insightful feedback. These comments have been valuable in the clarification of key aspects of our work and strengthen its presentation.
>
> **Assumption H2.** We have clarified in the main text the comparability of Assumption H2 with the standard estimation error assumption commonly used in the literature (see, e.g., Conforti et al., 2025). Notably, our main result remains valid if H2 is replaced by an $L^2$-type condition on the score function of the continuous-time forward process, combined with the requirement that $\tilde s_{\theta^\star}(T - t_k, \cdot)$ is uniformly Lipschitz in space. Importantly, Proposition B.2 ensures that this additional condition is not restrictive. Under these assumptions, we can apply the triangle inequality together with Proposition B.2 to obtain: $ ||\nabla \log \tilde p_{T-t_k}( X^\star_{t_k}) - \tilde s_{\theta^\star}( T-t_k, X^\star_{t_k}) || \leq || \nabla\log \tilde p_{T-t_k}(\overleftarrow X_{t_k} ) - \tilde s_{\theta^\star}( T-t_k,\overleftarrow X_{t_k}) || +(L+L')|| X^\star_{t_k}-\overleftarrow X_{t_k}||$,
> with $L'$ the Lipschitz constant of the score estimator. Replacing H2 with the standard assumption, the Lipschitz regularity of the score and its estimator combined with a straightforward generalisation of Proposition C.6 in (Strasman et al., 2024) would lead to only a minor adjustment in the definition of $\delta_k$ in Section D.3. As a result, aside from slight modifications to Lemma E.2, this substitution does not materially affect the proof of the main theorem or alter the key features of the final convergence bound. Hence, Assumption H2 is fully compatible with the classical framework, and its validity directly follows from the well-established correctness of the standard score estimation assumption. We would also like to refer to Appendix A in Chen et al. (2023), where it is demonstrated that the standard estimation error assumption holds in simple yet practically relevant scenarios. We opted for the formulation in Assumption H2 as it offers the most direct and tractable path within our proof technique. We hope that highlighting the connection between these formulations helps clarify the transparency of our bound and bridges the gap between different perspectives in the literature.
>
> **Link with the HJB equation.**
> We fully acknowledge the importance of the link with the Hamilton-Jacobi-Bellman (HJB) equation. This connection is rooted in a long history on the analysis of SGMs (see, e.g., Berner et al., 2022; Zhang and Katsoulakis, 2023; Zhang et al., 2024a; Conforti et al., 2025). Following the suggestion of the reviewer, we dedicated a discussion to clarify this connection explicitly in the revised version of the manuscript, showing the main works where this has been used.
>
> **Sentence constructions.**
> We acknowledge the ambiguity in the phrasing “not (necessarily) contractive” and thank the reviewer for pointing this out. In the revised version of the manuscript, we better specify this making reference to the Appendix where we define $T^\star$, the actual switching point in contractivity, and introduce $T(\alpha,M)$ as a lower bound on $T^\star$. Ultimately, this is not a limitation of our proof technique but rather a direct consequence of the fact that $T^\star$ is not explicitly computable.
>
> Finally, we have softened the original phrasing regarding the dependence on the dimension $d$. In the revised version of the paper, we now emphasize that our bound is in line with previously established results.
>
> Additionally, we would like to thank a reviewer for the misprint suggestion. We have happily introduced the required changes in the revised manuscript.

---

### Decision · Program_Chairs · 2025-05-01

**Decision:**

Accept (poster)

**Comment:**

The manuscript analyzes the Wasserstein distance between the data distribution and the generation from score-based models, based on the weak log-concavity assumption of the data distribution, which is tracked over the forward process through the analysis of the HJB equation for the log-density. Overall, the reviewers agree upon the novelty of the analysis for score-based generative models, the authors have addressed most of the concerns in the discussion phase. Hence the meta-reviewer recommends this paper for acceptance.